# Amulet: ReAlignment During Test Time for Personalized Preference Adaptation of LLMs

**Zhaowei Zhang**[1,2* ✉]    **Fengshuo Bai**[3,4*]    **Qizhi Chen**[1]    **Chengdong Ma**[1]
**Mingzhi Wang**[1]    **Haoran Sun**[1]    **Zilong Zheng**[2† ✉]    **Yaodong Yang**[1† ✉]

[1]Institute for Artificial Intelligence, Peking University
[2]State Key Laboratory of General Artificial Intelligence, BIGAI
[3]Shanghai Jiao Tong University
[4]Zhongguancun Academy

## Abstract

How to align large language models (LLMs) with user preferences from a static general dataset has been frequently studied. However, user preferences are usually personalized, changing, and diverse regarding culture, values, or time. This leads to the problem that the actual user preferences often do not coincide with those trained by the model developers in the practical use of LLMs. Since we cannot collect enough data and retrain for every demand, researching efficient real-time preference adaptation methods based on the backbone LLMs during test time is important. To this end, we introduce **Amulet**, a novel, training-free framework that formulates the decoding process of every token as a separate online learning problem with the guidance of simple user-provided prompts, thus enabling real-time optimization to satisfy users' personalized preferences. To reduce the computational cost brought by this optimization process for each token, we additionally provide a closed-form solution for each iteration step of the optimization process, thereby reducing the computational time cost to a negligible level. The detailed experimental results demonstrate that Amulet can achieve significant performance improvements in rich settings with combinations of different LLMs, datasets, and user preferences, while maintaining acceptable computational efficiency.

## 1 Introduction

The success of large language models (LLMs) has led to their widespread application in scenarios such as customer service (Raiaan et al., 2024), content creation (Hadi et al., 2024), and personal assistance (Chen et al., 2024), emphasizing the importance of maintaining alignment with human preferences (Ji et al., 2023; Anwar et al., 2024). However, existing LLM alignment researches often focus on aligning with user preferences from a *static* general dataset, neglecting *personalized* and changing preferences. This leads to challenges in ensuring that the alignment goals designed by model developers adequately address users' evolving needs in real-time post-deployment scenarios (Liao & Xiao, 2023; Lazar & Nelson, 2023; Zhang et al., 2024b; Corrêa, 2024; Zhang et al., 2024a).

One straightforward approach is to recollect data for personalized user preferences and use methods such as reinforcement learning from human feedback (RLHF) (Christiano et al., 2017; Ouyang et al., 2022) or direct preference optimization (DPO) (Rafailov et al., 2024) for further fine-tuning. However, these approaches not only face the difficulty in the requirements engineering problem of specifying users' real needs (Pohl, 1996; Mechergui & Sreedharan, 2024), but also from the fact that users' preferences continuously change with culture, community, context, scenario, and time (MacIntyre, 2013; Eckersley, 2018; Turchin, 2019; Zhang et al., 2023; Qiu et al., 2024; Zhu et al., 2025). If the requirements analysis, data collection, and subsequent fine-tuning processes are repeated each time, it will result in a significant cost burden. Just as shown in the (a) and (b) parts of the Figure 1, this phenomenon leads to a "last mile" problem (Boysen et al., 2021) in existing alignment research.

---

*Equal contribution.
†Corresponding author.

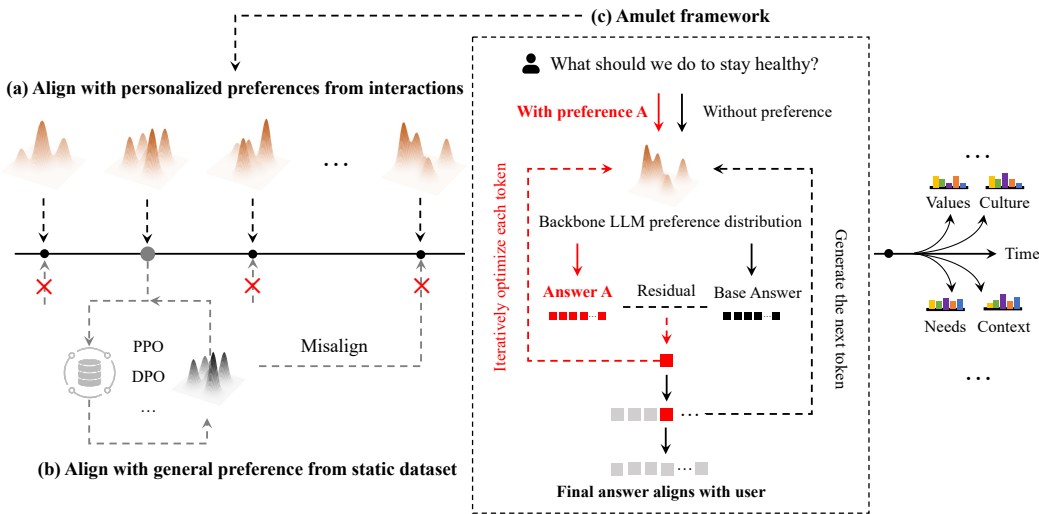

Figure 1: An illustration of our Amulet framework and its background. The figure is intersected by an axis, with each node on the axis displaying a different distribution that shows the constantly changing user personalized preferences due to factors like time, value, need, and context, as illustrated by the part (a). The part (b) shows that existing methods mostly consider aligning LLMs with general preferences from a static dataset, which may result in misalignment in dynamically personalized scenarios. In the part (c), we have enlarged one of the preference nodes to show the processing of our Amulet framework. We formulate the decoding process of every token as a separate online learning problem and further adapt the backbone LLMs to align with the current user preference through a real-time optimization process with the guidance of user-provided prompts. The red token means the current processing token, which will be the condition for the next token prediction.

To solve this problem, we believe a lightweight preference adaptation method implemented at the LLM test time is needed. Several existing works have already made some attempts in this regard. Assisted inference methods focus on training weak models to guide strong model inference in different aspects, including in natural language form (Ji et al., 2024; Bai et al., 2024), interpolation form (Liu et al., 2024b; Zheng et al., 2024), sampling form (Wu et al., 2024; Snell et al., 2024), representation form (Kong et al., 2024), and logits form (Mitchell et al., 2023; Liu et al., 2024a). RAIN (Li et al., 2023b) and URIAL (Lin et al., 2023) mainly focus on self-distillation, allowing the model to refine the tokens that better match user preferences continuously. Nevertheless, the efficiency of these methods is still not at an acceptable level for either the training of a weak model or the complex inference. Linear Alignment (LA) (Gao et al., 2024) proposes a linear approximation preference update method that achieves acceptable computational efficiency. However, it influences the inference process in only a basic manner, falling short of better approximating users' personalized preferences.

Therefore, in this paper, we suggest a new perspective that the problem can be solved by performing further online adaptation at test time based on the backbone LLMs. To this end, as illustrated in the (c) part of the Figure 1, we introduce **Amulet** (re**A**lign**M**ent d**U**ring test time for persona**L**ized pr**E**ference adap**T**ation), a novel, training-free framework that formulates the decoding process of every token as an independent online learning problem with the guidance of simple user-provided prompts, thus enabling real-time optimization to satisfy user preferences. Specifically, unlike methods such as RLHF (Christiano et al., 2017; Ouyang et al., 2022) that treat the entire decoding process as a Markov Decision Process (MDP) aiming to maximize cumulative reward, we consider iteratively optimizing the already-generated policies for each token's decoding process separately for more comprehensive preference approximation. We follow the idea of follow-the-regularized-leader (FTRL) (Hazan et al., 2016) to carry out the specific optimization process, and further improve the optimization process more stable in the LLM setting by introducing the proximal regularizer to the vanilla FTRL process (McMahan, 2011). We can then use the user-provided simple prompts to simulate the optimization direction by comparing whether there is a difference between policies given the prompts or not (Gao et al., 2024). Since this process may consume a significant amount of

computational cost, we further give a closed-form solution for each iteration, thereby reducing its computational and time costs to negligible levels.

In summary, our contributions are three-fold. **Firstly**, we suggest a new perspective that the generation of each token at test time can be modeled as an independent online optimization problem and provide the Amulet framework. This method can not only align with personalized and changing user preferences based on the backbone LLM policies but also does not require additional training and fine-tuning. To our knowledge, we are the first study to introduce the optimization process of online learning into the work of test-time preference alignment. **Secondly**, for each round of optimization iteration, we further provide its closed-form solution, thereby reducing the computational cost of iterative calculations to a negligible level. **Lastly**, we carried out comprehensive experiments to validate the efficacy of our framework across diverse settings, incorporating various combinations of LLMs, test datasets, and user preference dimensions. The findings reveal that our approach can achieve significant performance improvements and surpass all baseline methods in the majority of scenarios (an average of 75% among all the experimental settings, 100% for the best LLM, and 93.8% for the best preference), thus providing a final "amulet" for aligning personalized preferences.

## 2 RELATED WORK

In this section, we will introduce the background of the related research. The existing alignment methods for LLMs can generally be divided into three categories: training time alignment methods, assisted inference methods, and tuning-free methods. We will elaborate on them separately below.

**Alignment at Training Time.** This category is currently the most mainstream way of alignment, mainly focusing on first training the model itself on datasets, and then freezing the parameters for inference. There are many well-known algorithms, including RLHF (Christiano et al., 2017; Ouyang et al., 2022), CAI (Bai et al., 2022), DPO (Rafailov et al., 2024), ORPO (Hong et al., 2024), and KTO (Ethayarajh et al., 2024). In addition, there are some newer methods. SimPO (Meng et al., 2024) eliminates the dependence on a reference policy in DPO by introducing the sequence average log probability as an implicit reward. Quiet-STaR (Zelikman et al., 2024) enhances generalization across more reasoning tasks by training the model's reasoning and thinking abilities. Although these methods can achieve good results in capability metrics, they have to recollect enough data and perform further training and fine-tuning, making them unable to meet users' changing and personalized requirements in specific scenarios.

**Assisted Inference Methods.** This class of methods typically involves training a weak model (usually a small model or a pre-trained model) to enhance the alignment of a strong model during inference time, which has various implementation forms. Aligner (Ji et al., 2024) and Alignment via Bayesian Persuasion (Bai et al., 2024) use weak generated natural language to influence strong model behavior. ExPO (Zheng et al., 2024) adopts an interpolation approach, linearly combining the parameters of the small model with those of the large model to achieve alignment. Improved sampling strategies (Wu et al., 2024; Snell et al., 2024; Liu et al., 2025), from the perspective of optimal sampling, allow the strong model to generate higher-quality text under the guidance of a smaller reward model. EFT (Mitchell et al., 2023), DeRa (Liu et al., 2024b), and proxy-tuning (Liu et al., 2024a) integrate the logits of the aligned small model and the unaligned large model to guide the decoding process. RE-CONTROL (Kong et al., 2024) enhances the alignment ability of the large model by training a value model to edit the information representation of the large model during the decoding process. These methods reduce computation and time costs by training smaller models, but they are essentially the same as training-time methods and still fail to address the issue of changing and personalized preferences.

**Tuning-free Methods.** Methods in this class usually consider further optimization at inference time. RAIN (Li et al., 2023b) uses the LLM itself as a reward model to perform inference and rollback correction during the inference phase. URIAL (Lin et al., 2023) designs an inference approach by comparing the differences before and after model alignment, allowing the model to continuously correct and reinforce tokens that better match user preferences. However, both of these methods still require a significant amount of computational time and resources during inference. LA (Gao et al., 2024) proposes a method that compares the distribution changes caused by user preference prompts on the original model and linearly updates the original model's logits, which achieves acceptable computational efficiency. However, it influences the inference process in only a basic manner, falling short of better approximating users' preferences.

## 3 METHOD

In this section, we will formally introduce the Amulet framework. We begin by introducing our task settings, followed by reviewing online learning and the FTRL algorithm, and finally, we introduce the specific definition of our method and its closed-form solution.

### 3.1 TASK SETTINGS

Unlike most existing methods which view the decoding process of LLMs as an MDP aiming to maximize cumulative reward, our approach focuses on online optimization for the generation of each token at every timestep, thus enabling real-time optimization to satisfy user preferences.

For LLMs, the generation process of each token can be seen as a policy, representing the distribution of each token at the current moment. Typically, this policy is related to the model's parameters, but if we use only the already generated policy and perform post-processing on it, then it can be made independent of them. Therefore, our optimization target is the already generated policies for generating each token in the sequence.

Thus for the generation of each token, we need to find the following optimal policy $\pi^*(a)$:

$$\pi^*(a) = \arg\max_{\pi \in \Pi} \mathbb{E}_{a \sim \pi(\cdot|s,s_0)} r(a|s_0, s), \tag{1}$$

where $s_0$ represents the initial prompt, $s$ represents the sequence that has already been generated, $a \in \mathcal{A}$ denotes the optional token in the token space $\mathcal{A}$, and $r$ denotes the latent reward function that reflects the real current user preference.

### 3.2 FOLLOW-THE-REGULARIZED-LEADER ALGORITHM

We first briefly introduce online learning, which involves the training process of a model as it continuously receives information. Unlike common offline learning, it cannot access all the training data at once but can update iteratively, reflecting the impact of new situations in real time. This method allows the model to adapt to changing scenarios as well as explore and utilize unknown data distributions more effectively.

We will then review the FTRL algorithm, which is a frequently studied online learning framework (McMahan, 2011; Abe et al., 2022). Its core feature is the introduction of a strongly convex regularizer to the fictitious play process, which enhances the algorithm's stability and convergence (Hazan et al., 2016). Typically, the optimization process of FTRL for $\pi$ at the (t + 1)-th iteration can be expressed by the following iterative formula (Jacob et al., 2022; 2023):

$$\pi_{t+1}(a) = \arg\max_{\pi \in \Pi} \left[ \sum_{i=1}^{t} \mathcal{U}_i(\pi_i(a)) - \frac{\phi(\pi_t(a))}{\eta} \right]. \tag{2}$$

Because of the inability to obtain the user's true reward function, in each iteration $t$, we need an approximate utility function $\mathcal{U}_t$ to continuously provide an approximation of the user preferences for iteratively developing the policy that best meets user needs. The first item in the above formula is the fictitious play process, which aims to reduce the regret between the current policy and the historical expected policy, and the second one is the regularizer, $\eta > 0$ is the learning rate. Now, for each token's generation, we can adapt this iteration to post-process the policy and further get the optimal one.

### 3.3 AMULET FRAMEWORK

Based on the above introduction, we can see that for LLMs, the process of decoding each token at test time can be based on a customized utility function to achieve further optimization. Since we provide a general framework that is unrelated to the utility, the utility function only needs to reflect the relative quality of each token, and its selection can be very diverse, including methods based on inductive bias (Kadavath et al., 2022; Gao et al., 2024), human interaction (Mechergui & Sreedharan, 2024; Wang et al., 2024b), and environment feedback (Sutton, 2018; Le et al., 2022).

---

**Algorithm 1** Decoding Process with Amulet

---

**Require:** LLM for generating policy; basic prompt $p_{\text{base}}$; preference prompt $p_{\text{pref}}$; current generated
sequence $s$, iteration number $T$; maximum new token number $M$; parameters $\alpha$, $\lambda$, and $\eta$; blank
string $s$

1: **repeat**
2:     generate $\pi_1(a) = P_{\text{LLM}}(a|p_{\text{base}}, p_{\text{pref}}, s)$, $\pi_{\text{base}}(a) = P_{\text{LLM}}(a|p_{\text{base}}, s)$ with the given LLM
3:     **for** $t = 1, 2, \ldots, T - 1$ **do**
4:         calculate $u_t(\pi_t(a)) := \alpha(\log \pi_t(a) - \log \pi_{\text{base}}(a))$
5:         update the policy with the iteration given by Equation 6
6:     **end for**
7:     get the optimized policy $\pi^*(a) \leftarrow \pi_T(a)$
8:     sample the generated token $a$ with $\pi^*(a)$
9:     update the current sequence $s \leftarrow s + a$
10: **until** the length of $s$ reaches $M$ **or** generation is ended
11: **return** the full generation sequence $s$

---

Drawing from Contrastive Decoding (Li et al., 2022) and LA (Gao et al., 2024), we define the utility function to empirically simulate the optimization direction at time $t$ here as:

$$u_t(a) := \alpha(\log \pi_t(a) - \log \pi_{\text{base}}(a)). \tag{3}$$

Unlike recommendation systems that need to infer user preferences (Zhang et al., 2021), in our real-time setting, users directly provide explicit preferences in the form of prompts that reflect their current needs. Here, we define $p_{\text{base}}$ as the user's base prompts (e.g. questions), and $p_{\text{pref}}$ represents the user's specific real-time preferences prompts. $P_{\text{LLM}}(a|s)$ denotes the probability of generating each token $a$ by the LLM conditioned on the prompt $s$ at the current timestep. We then define the policy being optimized at the current moment $\pi_1(a) = P_{\text{LLM}}(a|p_{\text{base}}, p_{\text{pref}}, s)$ to simultaneously include $p_{\text{base}}$, $p_{\text{pref}}$, and the sequence of tokens $s$ generated at the current timestep. The base policy $\pi_{\text{base}}(a) = P_{\text{LLM}}(a|p_{\text{base}}, s)$ is a baseline policy that does not include user preferences, and $\alpha$ is an adjustable parameter. Therefore, the intuition of this utility is to gradually amplify the difference brought by the preference prompt $p_{\text{pref}}$ over the base prompt $p_{\text{base}}$ through a better policy during the iteration process and to further optimize in this direction for the current LLM, until the preference information it brings is fully exploited.

To avoid unreasonable optimization results, and to accelerate the convergence rate, we further introduce a KL regularization term into the utility function, which constrains the current policy $\pi(a)$ not to deviate too far from the initial one $\pi_1(a)$, and the ratio is adjusted by a controllable parameter $\lambda$. We define $u_t(\pi) = \langle u_t, \pi \rangle$, and therefore we can update the utility function as:

$$\mathcal{U}_t(\pi) := u_t(\pi) - \lambda D_{\text{KL}}(\pi \| \pi_1). \tag{4}$$

Typically, vanilla FTRL adopts an entropy item as the regularization term (Jacob et al., 2022). To make the optimization more stable, we introduce the KL term between $\pi(a)$ and $\pi_t(a)$ as the proximal convex regularizer. Subsequently, we will take Equation 4 into Equation 2, and we will obtain an FTRL-proximal-like (McMahan, 2011) iteration dynamics :

$$\pi_{t+1} = \arg\max_{\pi \in \Pi} \left[ \sum_{i=1}^{t} \mathcal{U}_i(\pi) - \frac{1}{\eta} D_{\text{KL}}(\pi \| \pi_t) \right]. \tag{5}$$

The dynamics, comprehensively exploiting the preference approximation brought about by the utility function, allows the policy to converge in the last iteration with a linear convergence rate. Detailed proofs are provided in Appendix A.2.

Next, we need to decide how to optimize this objective. Since our method requires a considerable number of optimization iterations for the generation of each token, it obviously consumes a lot of time and computational cost. Therefore, we further provide a closed-form solution for Equation 5, thereby reducing the computational cost of this iterative optimization to an almost negligible level.

**Proposition 3.1.** *The Equation 5 has a closed-form solution that is given by:*

$$\pi_{t+1}(a) \propto \exp\left(\frac{1}{t\lambda\eta + 1}\left(\eta\sum_{i=1}^{t} u_i(a) + \lambda\eta t \log\pi_1(a) + \log\pi_t(a)\right)\right). \qquad (6)$$

A complete derivation is provided in the Appendix A.1. We can follow this closed-form iteration to reach the optimal policy. If the number of iterations is fixed, the time complexity of our method is at the order of $O(n)$, where $n$ is the number of generated tokens. More computational efficiency details are provided in Appendix B.5. We have further provided the pseudo code for showing the details of the full decoding process with Amulet in Algorithm 1.

## 4 EXPERIMENTS

In this section, we conduct extensive experiments to evaluate Amulet with various combinations of LLMs, datasets, and user preferences. Our results demonstrate that our framework significantly improves LLMs' alignment performance, indicating its great potential for real-time user preference adaptation.

### 4.1 EXPERIMENT SETTINGS

We will first introduce the specific experimental setup, including the evaluated models and datasets, the baseline methods, and the evaluation metrics.

**Evaluated Models and Datasets.** In this paper, we evaluate four popular open-source models: Llama-2-7B-Chat (Touvron et al., 2023), Llama-3.1-8B-Instruct (Dubey et al., 2024), QWen2-7B-Instruct (Yang et al., 2024a;b), and Mistral-7B-Instruct-v0.2 (Jiang et al., 2023). These models were chosen for their diversity in architecture and performance characteristics, enabling a thorough assessment of our framework across different model types.

Since Amulet is designed without additional training or fine-tuning, we use the collected data solely for evaluation purposes. We construct four datasets for our experiments:

- **HelpSteer** (Wang et al., 2023) is a QA dataset aimed at evaluating the model's capability to follow instructions across five dimensions, including informativeness and factuality. We extracted the question part, focusing on single-sentence questions to create a dataset of 1,236 testing instances.

- **UltraFeedback** (Cui et al., 2023) is a comprehensive, high-quality AI feedback dataset designed to surpass traditional human feedback. From UltraFeedback, we selected two high-quality QA datasets: **Truthful QA** (Lin et al., 2021), which includes 811 testing problems, and **UltraChat** (Ding et al., 2023), from which we applied similar extraction and filtering as with HelpSteer, resulting in 3,845 testing problems.

- **Personal Preference Eval (Personal)** (Gao et al., 2024) is used to evaluate user preference alignment. We utilized the original dataset containing 548 testing instances.

For these datasets, we only use their questions, which is more similar to real-world applications where LLMs need to provide answers that align with users' real-time preferences for various questions. The extracted testing problems from these datasets serve as the base prompts for our experiment, providing a diverse range of user interactions to thoroughly evaluate the performance of the Amulet framework.

**Baseline Methods.** We compare the performance of our method with several baselines:

- **Base** refers to the original LLM using only the base prompt, which serves as the default response generation approach without any additional alignment or preference adjustment.

- **Preference (Pref)** involves the original LLM augmented with preference prompts for prompt engineering. The preference prompts are manually designed to reflect user preferences, serving as a way to enhance alignment without modifying the underlying model architecture.

Table 1: Results of our Amulet framework and all the other baselines on various combination settings of LLMs, user preferences, and datasets. All results are the arithmetic averages of the reward model scores on each dataset. The bold text in the table indicates the best performance under that setting. The colors in the table represent the percentage improvement of that method in the current setting relative to the Base method, with more positive growth bluer and more negative growth redder.

| Model | Dataset | Creative | | | | | Verbose | | | | | Concise | | | | | Uplifting | | | | |
|---|---|---|---|---|---|---|---|---|---|---|---|---|---|---|---|---|---|---|---|---|---|
| | | Base | Pref | BS16 | LA | Amulet | Base | Pref | BS16 | LA | Amulet | Base | Pref | BS16 | LA | Amulet | Base | Pref | BS16 | LA | Amulet |
| Mistral-7B | HelpSteer | 0.30 | 0.30 | 0.34 | 0.36 | **0.39** | 0.27 | 0.27 | **0.31** | 0.31 | 0.30 | 0.41 | 0.42 | 0.50 | 0.52 | **0.52** | 0.33 | 0.33 | 0.39 | 0.40 | **0.41** |
| | Personal | 0.34 | 0.34 | 0.35 | 0.38 | **0.42** | 0.30 | 0.30 | 0.30 | 0.30 | **0.30** | 0.47 | 0.49 | 0.50 | **0.54** | 0.53 | 0.41 | 0.42 | 0.42 | 0.45 | **0.46** |
| | Truthful QA | 0.32 | 0.33 | 0.34 | 0.38 | **0.41** | 0.30 | 0.31 | 0.31 | **0.33** | 0.32 | 0.41 | 0.44 | 0.47 | **0.51** | 0.49 | 0.36 | 0.38 | 0.39 | **0.47** | 0.47 |
| | Ultra Chat | 0.34 | 0.35 | 0.35 | 0.36 | **0.38** | 0.31 | 0.31 | 0.31 | **0.32** | 0.31 | 0.45 | 0.46 | 0.47 | 0.49 | **0.51** | 0.38 | 0.39 | 0.39 | 0.41 | **0.42** |
| | Average | 0.32 | 0.33 | 0.34 | 0.37 | 0.40 | 0.30 | 0.30 | 0.31 | 0.32 | 0.31 | 0.43 | 0.45 | 0.48 | 0.52 | 0.51 | 0.37 | 0.38 | 0.40 | 0.43 | 0.44 |
| Qwen2-7B | HelpSteer | 0.34 | 0.34 | 0.35 | 0.35 | **0.36** | 0.31 | 0.32 | **0.33** | 0.33 | 0.30 | 0.43 | 0.48 | 0.50 | 0.57 | **0.59** | 0.38 | 0.38 | 0.39 | 0.39 | **0.41** |
| | Personal | 0.33 | 0.34 | 0.34 | 0.37 | **0.41** | 0.31 | **0.31** | 0.31 | 0.30 | 0.28 | 0.41 | 0.48 | 0.49 | 0.53 | **0.54** | 0.40 | 0.42 | 0.42 | **0.43** | 0.42 |
| | Truthful QA | 0.32 | 0.33 | 0.33 | 0.34 | **0.36** | 0.30 | 0.31 | 0.32 | **0.33** | 0.32 | 0.41 | 0.46 | 0.50 | **0.54** | 0.51 | 0.36 | 0.38 | 0.39 | 0.44 | **0.45** |
| | Ultra Chat | 0.34 | 0.34 | 0.34 | 0.35 | **0.36** | 0.31 | 0.32 | 0.32 | **0.32** | 0.31 | 0.40 | 0.45 | 0.46 | 0.54 | **0.57** | 0.38 | 0.39 | 0.39 | **0.40** | 0.39 |
| | Average | 0.33 | 0.34 | 0.34 | 0.35 | 0.37 | 0.31 | 0.32 | 0.32 | 0.32 | 0.30 | 0.41 | 0.47 | 0.49 | 0.55 | 0.55 | 0.38 | 0.39 | 0.40 | 0.42 | 0.42 |
| Llama-3.1-8B | HelpSteer | 0.33 | 0.34 | 0.36 | 0.44 | **0.50** | 0.30 | 0.31 | 0.33 | 0.36 | **0.41** | 0.40 | 0.43 | 0.45 | 0.53 | **0.57** | 0.36 | 0.37 | 0.39 | 0.45 | **0.50** |
| | Personal | 0.35 | 0.36 | 0.36 | 0.46 | **0.62** | 0.31 | 0.31 | 0.31 | 0.35 | **0.49** | 0.39 | 0.44 | 0.45 | 0.53 | **0.67** | 0.42 | 0.44 | 0.43 | 0.49 | **0.61** |
| | Truthful QA | 0.31 | 0.33 | 0.33 | 0.41 | **0.56** | 0.29 | 0.29 | 0.31 | 0.34 | **0.44** | 0.37 | 0.40 | 0.42 | 0.49 | **0.52** | 0.34 | 0.36 | 0.37 | 0.43 | **0.49** |
| | Ultra Chat | 0.33 | 0.34 | 0.34 | 0.42 | **0.57** | 0.31 | 0.32 | 0.32 | 0.36 | **0.41** | 0.38 | 0.41 | 0.41 | 0.48 | **0.53** | 0.37 | 0.38 | 0.38 | 0.44 | **0.48** |
| | Average | 0.33 | 0.34 | 0.35 | 0.43 | 0.56 | 0.30 | 0.31 | 0.32 | 0.35 | 0.44 | 0.38 | 0.42 | 0.43 | 0.51 | 0.57 | 0.37 | 0.39 | 0.39 | 0.45 | 0.52 |
| Llama-2-7B | HelpSteer | 0.32 | 0.33 | 0.35 | **0.37** | 0.36 | 0.28 | 0.29 | 0.31 | **0.31** | 0.30 | 0.39 | 0.42 | 0.44 | **0.48** | 0.47 | 0.36 | 0.37 | 0.39 | **0.40** | 0.38 |
| | Personal | 0.32 | 0.33 | 0.32 | 0.39 | **0.45** | 0.26 | 0.27 | 0.27 | 0.29 | **0.32** | 0.38 | 0.41 | 0.43 | 0.49 | **0.53** | 0.40 | 0.41 | 0.41 | 0.45 | **0.49** |
| | Truthful QA | 0.30 | 0.32 | 0.31 | 0.36 | **0.41** | 0.27 | 0.28 | 0.28 | 0.30 | **0.32** | 0.30 | 0.35 | 0.37 | 0.44 | **0.49** | 0.34 | 0.36 | 0.36 | 0.40 | **0.44** |
| | Ultra Chat | 0.32 | 0.33 | 0.34 | 0.37 | **0.41** | 0.29 | 0.30 | 0.30 | 0.32 | **0.34** | 0.39 | 0.43 | 0.43 | 0.47 | **0.50** | 0.37 | 0.38 | 0.39 | 0.40 | **0.43** |
| | Average | 0.32 | 0.33 | 0.33 | 0.37 | 0.41 | 0.28 | 0.29 | 0.29 | 0.30 | 0.32 | 0.36 | 0.40 | 0.42 | 0.47 | 0.50 | 0.37 | 0.38 | 0.39 | 0.41 | 0.44 |

- **Beam Search (BS)** (Graves, 2012; Boulanger-Lewandowski et al., 2013) is a decoding strategy that helps LLMs find long-term optimal solutions by considering the square of the beam number of tokens during decoding. Due to its heavy memory consumption, we use a beam number of 16 (BS16) in this paper.
- **Linear Alignment (LA)** (Gao et al., 2024) is a token-level test-time alignment method that predicts the optimization direction of DPO and performs a linear update on the original policy. This is currently the state-of-the-art (SOTA) method for test-time personalized preference alignment.

**Evaluation Metrics.** Since our task aims to align open-ended user preferences, finding a targeted preference dataset to train the corresponding reward model as the metric is difficult. To address this issue, we selected the instruction-following dimension from ArmoRM-8B (Wang et al., 2024a) as the main evaluation metric for our experiment. This reward model is currently placed second in the RewardBench (Lambert et al., 2024) rankings [1].

To further ensure the completeness of the evaluation, we employed GPT-4o [2] as a discriminator to assess whether responses from different methods better met the requirements, categorizing outcomes as win, lose, or tie. The evaluation prompts were adapted from the AlpacaEval standard format (Li et al., 2023a) to fit our experimental context, with specific modifications detailed in the Appendix C.

### 4.2 EXPERIMENTAL RESULTS

To thoroughly and systematically evaluate our method's performance, we measure the average performance across different combinations of models, datasets, and preferences, as detailed in Table 1. Specifically, we consider eight preferences: creative, sycophantic, verbose, complex, formal, pleasant, concise, and uplifting. As indicated by prior research (Zhong et al., 2024), these preferences reflect common diverse user interaction scenarios and their demonstrated impact on user satisfaction.

The results show that our method significantly improves user preference alignment compared to all baseline methods measured by reward model score. In this section, we present the results for

---

[1] https://huggingface.co/spaces/allenai/reward-bench
[2] https://openai.com/index/hello-gpt-4o/

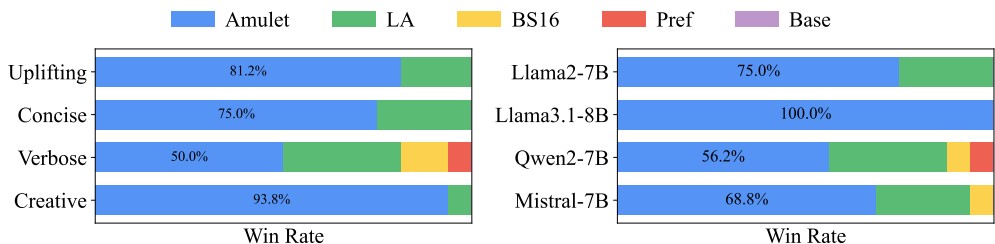

Figure 2: The percentage of the highest scores (win rate) for the 64 groups of experiments across all the methods with different user preferences and LLMs, measuring by the reward model metric.

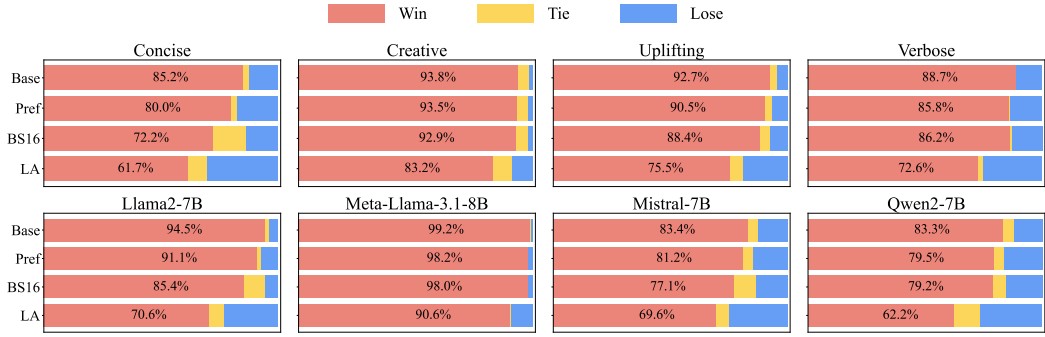

Figure 3: Detailed results on the GPT-4o win rate among Amulet versus all the other baselines (Base, Pref, BS16, and LA) on the Personal dataset. The first row of the figure shows the average win rate of Amulet for all the preferences and the second row for all the LLMs.

four representative preferences, while the results for the remaining four preferences are provided in Appendix B.1. We will then provide a detailed analysis of the experimental results from multiple perspectives.

**Overall Performance.** Table 1 presents the 64 groups of experiments we conducted with all methods across the combinations of 4 LLMs, 4 datasets, and 4 different user preferences. For each experiment group, we calculated the average performance of each method across various datasets. Our method surpassed all baseline methods in the majority of scenarios with a win rate (percentage of the highest scores) of 75%, reaching the current SOTA level. The following are LA (20%), BS16 (3%), Pref (2%), and Base (0%).

To provide a clearer and more direct comparison between different methods, we also calculated the win rates of all the methods across different user preferences and LLMs, which are shown in Figure 2. The results demonstrate that our method achieves the best performance compared to all baselines under all of the wide-range settings, showing strong versatility.

At the same time, we also conducted evaluation experiments using the GPT-4o win rate as a metric. Specifically, we used GPT-4o to judge whether the responses generated by two methods were more preferable (win), less preferable (lose), or equally preferable (tie) over the user preference. In this section, we conducted experiments on all four preferences and four LLMs using the Personal dataset, while more results are provided in Appendix B.2. Figure 3 shows the specific experimental results, where the first row of the figure shows the average win rate of Amulet for all the preferences and the second row shows the average win rate for all the LLMs. As shown in Figure 3, Amulet achieved the highest win rate in all tasks. Even the QWen2-7B model, which performed relatively weakly in Table 1, achieved a least win rate of 62.2%. More datasets and details are shown in Appendix B.3.

**Analysis for Different User Preferences.** We investigate the impact of different user preferences on the performance of our method. As shown in the left plot of Figure 2, from the win rate of our method in the 16 experiments conducted for each preference, the highest-ranking preference is creative, reaching 93.8%. Following that are uplifting (81.2%), concise (75%), and verbose (50%).

Table 2: Results of different model sizes. All the experiments were performed on the Personal dataset and evaluated by the reward model metric.

| Model | Creative | | | | | Verbose | | | | | Concise | | | | | Uplifting | | | | |
|---|---|---|---|---|---|---|---|---|---|---|---|---|---|---|---|---|---|---|---|---|
| | Base | Pref | BS16 | LA | **Amulet** | Base | Pref | BS16 | LA | **Amulet** | Base | Pref | BS16 | LA | **Amulet** | Base | Pref | BS16 | LA | **Amulet** |
| Qwen2-0.5B | 0.27 | **0.30** | 0.28 | 0.27 | 0.29 | 0.23 | **0.25** | 0.25 | 0.21 | 0.23 | 0.31 | 0.34 | 0.19 | 0.33 | **0.37** | 0.33 | 0.35 | 0.27 | 0.36 | **0.41** |
| Llama-3.2-1B | 0.28 | 0.27 | 0.34 | 0.34 | **0.35** | 0.23 | 0.22 | 0.30 | 0.33 | **0.36** | 0.31 | 0.29 | **0.40** | 0.38 | 0.39 | 0.34 | 0.32 | **0.41** | 0.39 | 0.38 |
| Llama-2-13B | 0.28 | 0.31 | 0.32 | 0.42 | **0.46** | 0.24 | 0.25 | 0.27 | 0.30 | **0.33** | 0.34 | 0.38 | 0.42 | 0.49 | **0.54** | 0.35 | 0.37 | 0.40 | 0.45 | **0.48** |
| Llama-2-70B | 0.33 | 0.33 | 0.33 | 0.39 | **0.43** | 0.28 | 0.28 | 0.28 | 0.32 | **0.33** | 0.44 | 0.51 | 0.50 | 0.57 | **0.62** | 0.40 | 0.41 | 0.41 | 0.44 | **0.47** |

Although the improvement on verbose was weaker than the other preferences, it still achieved a 50% win rate over all the other baselines, which demonstrates the effectiveness of our approach. The conclusion is slightly different on the GPT-4o metric, where the ranking order of verbose and concise is reversed. However, even for the concise preference, which ranks last, Amulet still has a 61.7% win rate against LA.

**Analysis for Different LLMs.** We also analyze the results in Table 1 from the perspective of LLM to highlight the performance improvements brought by our method to different models. Similarly, from the perspective of win rates, it can be seen that Llama-3.1-8B-Instruct achieved the best performance improvement, with a win rate of 100%. Its best-case scenario even shows a performance increase of 79% compared to the Base method, and a 35% improvement over the current SOTA method LA (creative preference at Personal dataset).

As shown in the right part of Figure 2, the following models are Llama-2-7B-Chat (75%), Mistral-7B-Instruct-v0.2 (68.8%), and QWen2-7B-Instruct (56.2%). Our method shows win rate improvements of these three models are 200%, 121%, and 49.9%, respectively, compared to the current SOTA method, LA. These experimental results demonstrate our method's effectiveness in enhancing the alignment of model responses with user preferences across various LLMs. This result also holds for the GPT-4o win rate metric. Moreover, as shown in the first row of Figure 3, Amulet even shows better performance on the GPT-4o win rate metric compared with the results in Table 1, achieving a 62.2% win rate against LA on the QWen2-7B model.

## 4.3 ABLATION STUDIES

In this section, we conduct a more comprehensive analysis to study the effectiveness of our method in a wider range of scenarios with various model sizes and the impact of different parameters on performance. More ablation study details are presented in the Appendix B.4.

**Analysis for Different Model Sizes.** To ensure the comprehensiveness of our study, we also conducted additional experiments to analyze the impact of different model sizes on the performance of our method. Specifically, we have added experiments with two small models, Qwen2-0.5B-Instruct (Yang et al., 2024a;b) and Llama-3.2-1B-Instruct (Meta, 2024), and two big models, Llama-2-13B-Chat and Llama-2-70B-Chat (Touvron et al., 2023). All the experiments were conducted on the Personal dataset. As illustrated in Table 2, Amulet also demonstrates excellent performance across different model sizes. For Llama-2-13B-Chat and Llama-2-70B-Chat, Amulet achieved the best performance in all four preferences. For the small models Qwen2-0.5B-Instruct and Llama-3.2-1B-Instruct, Amulet achieved the best performance in half of the tasks. This demonstrates that Amulet is still able to perform excellently on models of different sizes.

**Analysis for Parameter Settings.** We then conduct experiments to study the impact of various parameter settings on the performance of our method. We fixed all other parameters and analyzed each parameter individually. We will introduce them one by one as follows:

- **Iteration Number $T$.** We conduct experiments using 0, 20, 40, 60, 80, and 100 iterations. As shown in Figure 4, the results indicate that increasing the number of iterations generally improves performance on both metrics. Notably, a significant improvement is observed between 0 and 20 iterations, while most runs achieve optimal performance between 40 and 80 iterations. Interestingly, too many iterations lead to a decline in performance. We suppose that may be attributed to the over-fitting of the utility which is only an approximation of the latent real preference.
- **Learning Rate $\eta$.** We conduct the experiments ranging from 2, 4, . . . , 20. As shown in the first subplot of Figure 5, as the learning rate $\eta$ increases, the performance initially rises sharply, then

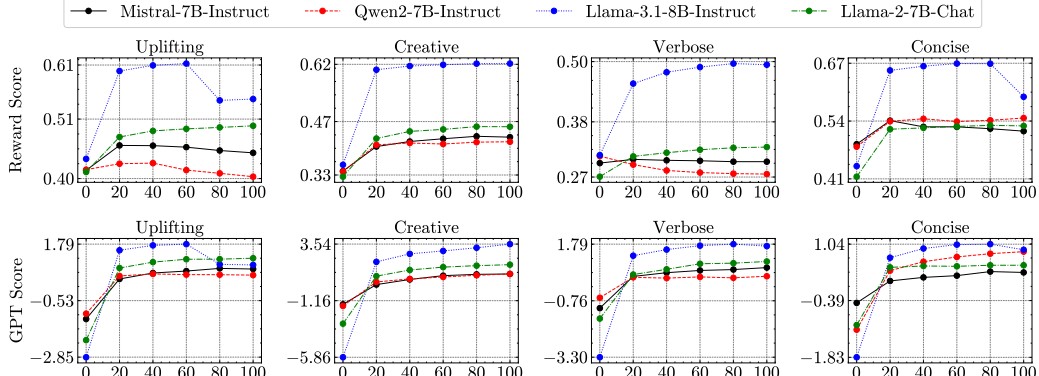

Figure 4: Effect of Iteration Number. The experiments are conducted using different iteration numbers on the Personal dataset, involving four distinct LLMs and various user preference dimensions. The evaluation metrics, presented in the first and second rows of the figure, are the reward model score and the GPT-4o based Bradley–Terry (BT) score (Bradley & Terry, 1952; Ouyang et al., 2022; Rafailov et al., 2024), respectively.

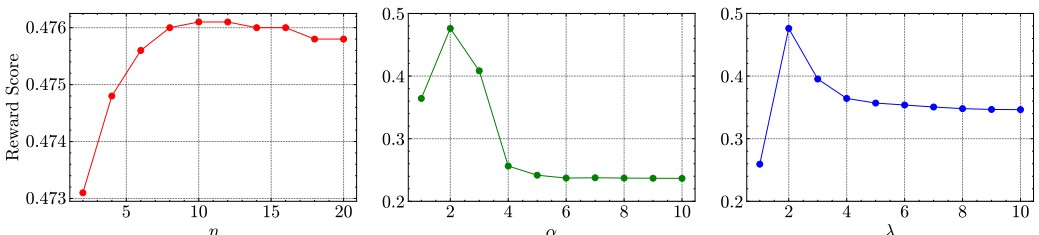

Figure 5: Effect of different $\eta$, $\alpha$, and $\lambda$ values on the Truthful QA dataset using Llama-3.1-8B-Instruct for the creative preference dimension. The evaluation metric is the reward model score.

slowly declines, and eventually stabilizes. Therefore, we chose $\eta = 10$ as our final parameter, where the turning point is observed.

- **Parameter $\alpha$ and $\lambda$.** In Equation 4, $\alpha$ and $\lambda$ are key parameters for adjusting the utility function: $\alpha$ regulates the balance between approximating user preferences, while $\lambda$ constrains the current policy to avoid significant deviations from the initial policy. We conduct experiments of both the parameters ranging from 1, 2, . . . , 10. As shown in the last two subplots of Figure 5, both parameters exhibit a similar trend: performance rises rapidly from 0 to 2, declines from 2 to 4, and then stabilizes. Based on the experimental results, we set both parameters to 2 to balance performance and stability.

More details of the analysis for our parameter settings can be found in Appendix B.4.

## 5 CONCLUSION

In this work, we introduce Amulet, a novel framework that formulates the decoding process of each token as an online learning problem, thus enabling real-time optimization to satisfy users' evolving personalized preferences. To alleviate the computational cost brought by the optimization process for each token, we further provide a closed-form solution for each iteration, thereby reducing the computational and time cost to a negligible level. We conducted extensive experiments to assess the effectiveness of our framework in a wide range of settings, including different combinations of LLMs, test datasets, and user preference dimensions. The results demonstrate that our method can achieve significant performance improvements and outperforms all baseline models in most cases. To our knowledge, we are the first to introduce the optimization process of online learning into the work of test-time preference alignment. Our work not only provides a valuable method but also offers a novel framework and perspective. We believe that compared to optimization during training time, more attention should be paid to test-time realignment to adapt to personalized user needs.

## ACKNOWLEDGMENTS

The work of Amulet was supported by the National Natural Science Foundation of China (62376031).

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

# Supplementary Material

## Table of Contents

# A   DETAILED THEORETICAL RESULTS

In this section, we will provide specific proof for the theoretical properties of our method.

## A.1   DERIVATION OF THE CLOSED-FORM SOLUTION

We try to solve the close-form solution about Equation 5:

$$\pi_{t+1} = \arg\max_{\pi \in \Pi} \left[ \sum_{i=1}^{t} \mathcal{U}_i(\pi) - \frac{1}{\eta} D_{\mathrm{KL}}(\pi \| \pi_t) \right]. \tag{5}$$

consider to maximize the following objective function:

$$\mathcal{L}(\pi_{t+1}, \mu) = \underbrace{\sum_{i=1}^{t} \sum_{a \in A} \pi_t(a) u_i(a)}_{(1)} - \underbrace{\sum_{i=1}^{t} \lambda(\pi_{t+1}(a) \log \frac{\pi_{t+1}(a)}{\pi_1(a)})}_{(2)}$$

$$- \underbrace{\frac{1}{\eta} \sum_{a \in A} \pi_{t+1}(a) \log \frac{\pi_{t+1}(a)}{\pi_t(a)}}_{(3)} + \underbrace{\sum_{i=1}^{t+1} \mu_i \left( 1 - \sum_{a \in A} \pi_i(a) \right)}_{(4)}$$

Here, (1) and (2) originate from the utility function $\mathcal{U}$, (3) from the KL divergence, and the in (4), $\mu_i$ is the Lagrange multiplier. We calculate the derivation of the function for a given $a$, we have

$$\frac{\partial \mathcal{L}(\pi_{t+1}, \mu)}{\partial \pi_{t+1}(a)} = \sum_{i=1}^{t} u_i(a) - t\lambda \left( \log \frac{\pi_{t+1}(a)}{\pi_1(a)} + 1 \right) - \frac{1}{\eta} \left( \log \frac{\pi_{t+1}(a)}{\pi_t(a)} + 1 \right) - \mu_{t+1}$$

Rearrange the terms:

$$\sum_{i=1}^{t} u_i(a) - t\lambda \log \pi_{t+1}(a) + t\lambda \log \pi_1(a) - \frac{1}{\eta} \log \pi_{t+1}(a) + \frac{1}{\eta} \log \pi_t(a) - t\lambda - \frac{1}{\eta} - \mu_{t+1} = 0$$

Combine the coefficients of $\log \pi_{t+1}(a)$:

$$-(t\lambda + \frac{1}{\eta}) \log \pi_{t+1}(a) = -\sum_{i=1}^{t} u_i(a) - t\lambda \log \pi_1(a) - \frac{1}{\eta} \log \pi_t(a) + t\lambda + \frac{1}{\eta} + \mu_{t+1}$$

Solve for $\log \pi_{t+1}(a)$:

$$\log \pi_{t+1}(a) = \frac{1}{t\lambda + \frac{1}{\eta}} \left( \sum_{i=1}^{t} u_i(a) + t\lambda \log \pi_1(a) + \frac{1}{\eta} \log \pi_t(a) - t\lambda - \frac{1}{\eta} - \mu_{t+1} \right)$$

Simplify the constant term, let

$$C_3 = -\frac{t\lambda + \frac{1}{\eta} + \mu_{t+1}}{t\lambda + \frac{1}{\eta}}$$

we have:

$$\log \pi_{t+1}(a) = \frac{1}{t\lambda + \frac{1}{\eta}} \left( \sum_{i=1}^{t} u_i(a) + t\lambda \log \pi_1(a) + \frac{1}{\eta} \log \pi_t(a) \right) + C_3$$

This is equivalent to the following expression:

$$\pi_{t+1}(a) \propto \exp\left(\frac{1}{t\lambda\eta + 1}\left(\eta\sum_{i=1}^{t} u_i(a) + \lambda\eta t \log \pi_1(a) + \log \pi_t(a)\right)\right)$$

## A.2 CONVERGENCE OF THE ALGORITHM

In this section, we aim to prove the convergence of our algorithm by analyzing the behavior of the KL divergence over iterations. The key idea is to show that the KL divergence between the optimal distribution $\pi^*$ and the current iterate $\pi_{t+1}$ decreases geometrically, leading to convergence. For convenient, we write $u_*(\pi')$ as $\pi_t = \pi^*$ in the utility function.

*Proof.* We begin by defining the update rule for the algorithm:

$$T(y_t) = \arg\max \langle y_t, \pi \rangle - \psi(\pi) \tag{7}$$

where

$$y_t = \eta\sum_{i=1}^{t-1}\nabla_\pi \mathcal{U}_i(\pi) = \eta\sum_{i=1}^{t-1}(\nabla_\pi u_i(\pi) - \lambda\nabla_\pi D_{\mathrm{KL}}(\pi||\pi_1))$$

and $\psi(\pi) = D_{\mathrm{KL}}(\pi||\pi_{t-1})$

To facilitate the analysis, we introduce the following equations and lemmas:

**Lemma 1.**

$$\left\langle\log\frac{\rho}{\pi}, \pi^* - \rho\right\rangle = -D_{\mathrm{KL}}(\pi^*||\rho) + D_{\mathrm{KL}}(\pi^*||\pi) - D_{\mathrm{KL}}(\rho||\pi)$$

This can be proven by directly expanding the terms.

**Lemma 2.** $D_\phi(\pi_t, T(y_t)) = \phi(\pi_t) - \phi(T(y_t)) - \langle y_i, \pi_i - T(y_i)\rangle$

The prove are following.

*Proof.* By definition, the Bregman divergence is given by:

$$D_\phi(\pi_i, T(y_i)) = \phi(\pi_i) - \phi(T(y_i)) - \langle\nabla\phi(T(y_i)), \pi_i - T(y_i)\rangle$$

Since $\pi$ is a probability distribution, it satisfies the linear constraint $Ax = b$. Using the Lagrange method, we define the Lagrangian:

$$\mathcal{L}(x, \nu) = \langle y_i, x\rangle - \phi(x) + \nu^T(b - Ax)$$

To find the stationary points, we solve for the gradient:

$$\nabla\mathcal{L}(x^*, v) = y_i - \nabla\phi(x^*) - A^T\nu = 0$$

Thus, we obtain:

$$\begin{aligned}
\langle y_i, \pi_i - T(y_i)\rangle &= \langle\nabla\phi(T(y_i)), \pi_i - T(y_i)\rangle + \nu^T A\pi_i - \nu^T A(y_i)\\
&= \langle\nabla\phi(T(y_i)), \pi_i - T(y_i)\rangle + \nu^T b - \nu^T b\\
&= \langle\nabla\phi(T(y_i)), \pi_i - T(y_i)\rangle
\end{aligned}$$

This completes the proof. □

**Lemma 3.** *Define* $\psi(\pi) = D_{\mathrm{KL}}(\pi||\pi_t) = \sum_a \pi(a)\log\frac{\pi(a)}{\pi_t(a)}$. *for any distributions $a$ and $b$, we have:*

$$D_\psi(a, b) = D_{\mathrm{KL}}(a||b)$$

*Proof.* By the definition of the Bregman divergence, we have:

$$D_\psi(\pi^*||\pi_{t+1}) = \psi(\pi^*) - \psi(\pi_{t+1}) - \left\langle \nabla_{\pi_{t+1}} \psi(\pi_{t+1}), \pi^* - \pi_{t+1} \right\rangle$$

Substituting the expression for $\psi$, we obtain:

$$D_\psi(\pi^*||\pi_{t+1}) = D_{\mathrm{KL}}(\pi^*||\pi_t) - D_{\mathrm{KL}}(\pi_{t+1}||\pi_t) - \left\langle \nabla_{\pi_{t+1}} D_{\mathrm{KL}}(\pi_{t+1}||\pi_t), \pi^* - \pi_{t+1} \right\rangle$$

By simplifying these terms, we arrive at:

$$D_\psi(\pi^*||\pi_{t+1}) = D_{\mathrm{KL}}(\pi^*||\pi_{t+1})$$

This shows that the Bregman divergence $D_\psi(\pi^*, \pi_{t+1})$ is equivalent to the KL divergence $D_{\mathrm{KL}}(\pi^*||\pi_{t+1})$. □

Using Lemma 3, we begin by examining the relationship between the KL divergences at consecutive iterations:

$$D_{\mathrm{KL}}(\pi^*||\pi_{t+1}) - D_{\mathrm{KL}}(\pi^*||\pi_t) + D_{\mathrm{KL}}(\pi_{t+1}||\pi_t) = D_\psi(\pi^*, \pi_{t+1}) - D_\psi(\pi^*, \pi_t) + D_\psi(\pi_{t+1}, \pi_t)$$

This can be expanded as:

$$
\begin{aligned}
&= \psi(\pi^*) - \psi(\pi_{t+1}) - \langle y_t, \pi^* - \pi_{t+1}\rangle - \psi(\pi^*) + \psi(\pi_t) \\
&\quad\quad + \langle y_{t-1}, \pi^* - \pi_t \rangle + \psi(\pi_{t+1}) - \psi(\pi_t) - \langle y_{t-1}, \pi_{t+1} - \pi_t \rangle \\
&= \langle y_t - y_{t-1}, \pi_{t+1} - \pi^* \rangle \\
&= \eta \left\langle \nabla_{\pi_t} u_t(\pi^t) - \lambda \nabla_{\pi_t} D_{\mathrm{KL}}(\pi_t||\pi_1), \pi_{t+1} - \pi^* \right\rangle
\end{aligned}
$$

Thus, we obtain the inequality:

$$D_{\mathrm{KL}}(\pi^*||\pi_{t+1}) - D_{\mathrm{KL}}(\pi^*||\pi_t) + D_{\mathrm{KL}}(\pi_{t+1}||\pi_t) \leq \eta \left\langle \nabla_{\pi_t} u_t(\pi_t) - \lambda \nabla_{\pi_t} D_{\mathrm{KL}}(\pi_t||\pi_1), \pi_{t+1} - \pi^* \right\rangle$$

Next, we consider the second term:

$$
\begin{aligned}
&- \left\langle \lambda \nabla_{\pi_t} D_{\mathrm{KL}}(\pi_t||\pi_1), \pi_{t+1} - \pi^* \right\rangle \\
&= \left\langle \lambda \nabla_{\pi_t} D_{\mathrm{KL}}(\pi_t||\pi_1), \pi_t - \pi_{t+1} \right\rangle + \left\langle \lambda \nabla_{\pi_t} D_{\mathrm{KL}}(\pi_t||\pi_1), \pi^* - \pi_t \right\rangle
\end{aligned}
$$

We then analyze:

$$
\begin{aligned}
&\left\langle \lambda \nabla_{\pi_t} D_{\mathrm{KL}}(\pi_t||\pi_1), \pi_t - \pi_{t+1} \right\rangle \\
\leq &D_{\mathrm{KL}}(\pi_t||\pi_1) - D_{\mathrm{KL}}(\pi_{t+1}||\pi_1) + D_{\mathrm{KL}}(\pi_{t+1}||\pi_t) \\
\leq &D_{\mathrm{KL}}(\pi_t||\pi_1) - D_{\mathrm{KL}}(\pi^*||\pi_1) + \left\langle \nabla_{\pi^*} D_{\mathrm{KL}}(\pi^*||\pi_1), \pi^* - \pi_{t+1} \right\rangle + D_{\mathrm{KL}}(\pi_{t+1}||\pi_t) \\
\leq &\left\langle \nabla_{\pi_t} D_{\mathrm{KL}}(\pi_t||\pi_1), \pi_t - \pi^* \right\rangle - D_{\mathrm{KL}}(\pi^*||\pi_t) \\
&\quad\quad + \left\langle \nabla_{\pi^*} D_{\mathrm{KL}}(\pi^*||\pi_1), \pi^* - \pi_{t+1} \right\rangle + D_{\mathrm{KL}}(\pi_{t+1}||\pi_t)
\end{aligned}
$$

The first inequality arises from:

$$D_{\mathrm{KL}}(\pi_{t+1}||\pi_1) \geq D_{\mathrm{KL}}(\pi^*||\pi_1) + \left\langle \nabla_{\pi^*} D_{\mathrm{KL}}(\pi^*||\pi_1), \pi_{t+1} - \pi^* \right\rangle$$

And the second equality follows from:

$$D_{\mathrm{KL}}(\pi^*||\pi_1) - D_{\mathrm{KL}}(\pi_t||\pi_1) = \left\langle \nabla_{\pi_t} D_{\mathrm{KL}}(\pi_t||\pi_1), \pi^* - \pi_t \right\rangle + D_{\mathrm{KL}}(\pi^*||\pi_t)$$

Combining these, we have:

$$- \langle \lambda \nabla_{\pi_t} D_{\mathrm{KL}}(\pi_t || \pi_1), \pi_{t+1} - \pi^* \rangle$$
$$\leq D_{\mathrm{KL}}(\pi_{t+1} || \pi_t) - D_{\mathrm{KL}}(\pi^* || \pi_t) + \langle \nabla_{\pi^*} D_{\mathrm{KL}}(\pi^* || \pi_1), \pi^* - \pi_{t+1} \rangle$$

Thus,

$$D_{\mathrm{KL}}(\pi^* || \pi_{t+1}) - D_{\mathrm{KL}}(\pi^* || \pi_t) + D_{\mathrm{KL}}(\pi_{t+1} || \pi_t)$$
$$\leq \eta \lambda D_{\mathrm{KL}}(\pi_{t+1} || \pi_t) - \eta \lambda D_{\mathrm{KL}}(\pi^* || \pi_t) + \eta \lambda \langle \nabla_{\pi^*} D_{\mathrm{KL}}(\pi^* || \pi_1), \pi^* - \pi_{t+1} \rangle$$
$$+ \eta \langle \nabla_{\pi_t} u_t(\pi_t), \pi_{t+1} - \pi^* \rangle$$

we rearrange the terms,

$$D_{\mathrm{KL}}(\pi^* || \pi_{t+1}) - (1 - \eta \lambda) D_{\mathrm{KL}}(\pi^* || \pi_t) + (1 - \eta \lambda) D_{\mathrm{KL}}(\pi_{t+1} || \pi_t)$$
$$\leq \eta \lambda \langle \nabla_{\pi^*} D_{\mathrm{KL}}(\pi^* || \pi_1), \pi^* - \pi_{t+1} \rangle + \eta \langle \nabla_{\pi_t} u_t(\pi_t), \pi_{t+1} - \pi^* \rangle$$

As we expand further, it becomes evident how the utility gradients at consecutive time steps contribute to the policy update:

$$D_{\mathrm{KL}}(\pi^* || \pi_{t+1}) - (1 - \eta \lambda) D_{\mathrm{KL}}(\pi^* || \pi_t) + (1 - \eta \lambda) D_{\mathrm{KL}}(\pi_{t+1} || \pi_t)$$
$$\leq \eta \lambda \langle \nabla_{\pi^*} D_{\mathrm{KL}}(\pi^* || \pi_1), \pi^* - \pi_{t+1} \rangle + \eta \langle \nabla_{\pi_t} u_t(\pi_t), \pi_{t+1} - \pi^* \rangle$$
$$\leq \eta \lambda \langle \nabla_{\pi^*} D_{\mathrm{KL}}(\pi^* || \pi_1), \pi^* - \pi_{t+1} \rangle + \eta \langle \nabla_{\pi_{t+1}} u_{t+1}(\pi_{t+1}), \pi_{t+1} - \pi^* \rangle$$
$$+ \eta \langle \nabla_{\pi_t} u_t(\pi_t) - \nabla_{\pi_t} u_{t+1}(\pi_{t+1}), \pi_{t+1} - \pi^* \rangle$$
$$\leq \eta \langle \nabla_{\pi_{t+1}} u_{t+1}(\pi_{t+1}) - \lambda \nabla_{\pi^*} D_{\mathrm{KL}}(\pi^* || \pi_1) \pi_{t+1} - \pi^* \rangle$$
$$+ \eta \langle \nabla_{\pi_t} u_t(\pi_t) - \nabla_{\pi_t} u_{t+1}(\pi_{t+1}), \pi_{t+1} - \pi^* \rangle$$
$$\leq \eta \langle \nabla_{\pi^*} u_*(\pi^*) - \lambda \nabla_{\pi^*} D_{\mathrm{KL}}(\pi^* || \pi_1), \pi_{t+1} - \pi^* \rangle$$
$$+ \eta \langle \nabla_{\pi_t} u_t(\pi_t) - \nabla_{\pi_t} u_{t+1}(\pi_{t+1}), \pi_{t+1} - \pi^* \rangle$$
$$\leq \eta \langle \nabla_{\pi_t} u_t(\pi_t) - \nabla_{\pi_t} u_{t+1}(\pi_{t+1}), \pi_{t+1} - \pi^* \rangle$$

where we use that

$$\langle \nabla_\pi u_*(\pi^*) - \nabla_\pi u_{t+1}(\pi_{t+1}), \pi^* - \pi_{t+1} \rangle \leq 0$$

and an inequality derived from the first-order optimality condition

$$\langle \nabla_{\pi^*} u_*(\pi^*) - \lambda \nabla_{\pi^*} D_{\mathrm{KL}}(\pi^* || \pi_1), \pi^* - \pi_{t+1} \rangle \geq 0$$

Considering a specific form for the utility function, $u_t(\pi_t) = \log \pi_t - \log \pi_{\mathrm{base}}$, using Lemma 1 again, we refine the inequality further:

$$D_{\mathrm{KL}}(\pi^* || \pi_{t+1}) - (1 - \eta \lambda) D_{\mathrm{KL}}(\pi^* || \pi_t) + (1 - \eta \lambda) D_{\mathrm{KL}}(\pi_{t+1} || \pi_t)$$
$$\leq \eta D_{\mathrm{KL}}(\pi^* || \pi_{t+1}) - \eta D_{\mathrm{KL}}(\pi^* || \pi_t) + \eta D_{\mathrm{KL}}(\pi_{t+1} || \pi_t)$$

That is

$$D_{\mathrm{KL}}(\pi^* || \pi_{t+1}) \leq (1 - \eta \lambda - \eta) D_{\mathrm{KL}}(\pi^* || \pi_t) - (1 - \eta \lambda - \eta) D_{\mathrm{KL}}(\pi_{t+1} || \pi_t)$$

Finally, under the condition $1 > 1 - \eta \lambda - \eta > 0$, we will get

$$D_{\mathrm{KL}}(\pi^* || \pi_{t+1}) \leq (1 - \eta \lambda - \eta) D_{\mathrm{KL}}(\pi^* || \pi_t) \leq (1 - \eta \lambda - \eta)^t D_{\mathrm{KL}}(\pi^* || \pi_1)$$

This shows that the policy updates effectively lead to convergence towards the optimal policy $\pi^*$.

$\square$

Table 3: Results of our Amulet framework and all the other baselines on various combination settings of models, user preferences, and test datasets. All results are the arithmetic averages of the reward model scores on each test set. The bold text in the table indicates the best performance under that setting. The colors in the table represent the percentage improvement of that method in the current setting relative to the Base method, with more positive growth bluer and more negative growth redder.

| Model | Dataset | Sycophantic | | | | | Formal | | | | | Pleasant | | | | | Complex | | | | |
|---|---|---|---|---|---|---|---|---|---|---|---|---|---|---|---|---|---|---|---|---|---|
| | | Base | Pref | BS16 | LA | Amulet | Base | Pref | BS16 | LA | Amulet | Base | Pref | BS16 | LA | Amulet | Base | Pref | BS16 | LA | Amulet |
| Mistral-7B | HelpSteer | 0.32 | 0.32 | 0.38 | 0.49 | **0.52** | 0.43 | 0.44 | **0.53** | 0.52 | 0.50 | 0.40 | 0.40 | **0.47** | 0.47 | 0.46 | 0.32 | 0.32 | **0.38** | 0.37 | 0.37 |
| | Personal | 0.35 | 0.37 | 0.36 | 0.57 | **0.59** | 0.52 | 0.53 | 0.53 | **0.56** | 0.54 | 0.46 | 0.47 | 0.47 | **0.48** | 0.47 | 0.35 | 0.36 | 0.35 | 0.37 | **0.38** |
| | Truthful QA | 0.33 | 0.35 | 0.35 | 0.54 | **0.55** | 0.52 | 0.53 | 0.55 | **0.59** | 0.56 | 0.45 | 0.47 | 0.47 | **0.51** | 0.49 | 0.38 | 0.39 | 0.39 | **0.41** | 0.39 |
| | Ultra Chat | 0.36 | 0.36 | 0.38 | 0.50 | **0.54** | 0.50 | 0.51 | 0.52 | **0.52** | 0.51 | 0.46 | 0.46 | 0.47 | **0.47** | 0.47 | 0.38 | 0.38 | 0.38 | **0.38** | 0.38 |
| | Average | 0.34 | 0.35 | 0.37 | 0.52 | 0.55 | 0.49 | 0.50 | 0.53 | 0.55 | 0.53 | 0.44 | 0.45 | 0.47 | 0.48 | 0.47 | 0.36 | 0.36 | 0.38 | 0.38 | 0.38 |
| Qwen2-7B | HelpSteer | 0.36 | 0.37 | 0.38 | 0.47 | **0.50** | 0.49 | 0.51 | 0.52 | **0.54** | 0.51 | 0.45 | 0.46 | **0.46** | 0.46 | 0.45 | 0.38 | 0.39 | 0.40 | **0.40** | 0.37 |
| | Personal | 0.36 | 0.37 | 0.36 | 0.46 | **0.46** | 0.51 | 0.52 | 0.53 | **0.55** | 0.53 | 0.46 | 0.46 | 0.47 | 0.46 | **0.47** | 0.37 | 0.38 | 0.38 | **0.39** | 0.33 |
| | Truthful QA | 0.34 | 0.36 | 0.36 | 0.44 | **0.46** | 0.52 | 0.54 | 0.56 | **0.57** | 0.53 | 0.44 | 0.45 | 0.46 | **0.46** | 0.45 | 0.39 | 0.41 | 0.40 | **0.42** | 0.37 |
| | Ultra Chat | 0.36 | 0.38 | 0.38 | 0.47 | **0.49** | 0.49 | 0.51 | 0.52 | **0.53** | 0.51 | 0.45 | 0.45 | **0.46** | 0.45 | 0.44 | 0.39 | 0.40 | 0.40 | **0.41** | 0.36 |
| | Average | 0.36 | 0.37 | 0.37 | 0.46 | 0.48 | 0.50 | 0.52 | 0.53 | 0.55 | 0.52 | 0.45 | 0.46 | 0.46 | 0.46 | 0.45 | 0.38 | 0.40 | 0.40 | 0.40 | 0.36 |
| Llama-3.1-8B | HelpSteer | 0.34 | 0.34 | 0.37 | 0.49 | **0.57** | 0.46 | 0.47 | 0.51 | 0.52 | **0.53** | 0.43 | 0.43 | 0.47 | 0.51 | **0.53** | 0.36 | 0.36 | 0.39 | 0.40 | **0.42** |
| | Personal | 0.34 | 0.36 | 0.36 | 0.55 | **0.69** | 0.50 | 0.52 | 0.52 | 0.55 | **0.61** | 0.48 | 0.49 | 0.48 | 0.53 | **0.62** | 0.37 | 0.38 | 0.37 | 0.40 | **0.52** |
| | Truthful QA | 0.32 | 0.32 | 0.34 | 0.45 | **0.63** | 0.47 | 0.48 | 0.53 | 0.53 | **0.55** | 0.42 | 0.43 | 0.45 | 0.49 | **0.52** | 0.36 | 0.37 | 0.40 | 0.41 | **0.50** |
| | Ultra Chat | 0.34 | 0.35 | 0.36 | 0.48 | **0.56** | 0.48 | 0.49 | 0.50 | 0.52 | **0.53** | 0.44 | 0.45 | 0.45 | 0.49 | **0.52** | 0.38 | 0.39 | 0.39 | 0.41 | **0.42** |
| | Average | 0.34 | 0.34 | 0.36 | 0.49 | 0.61 | 0.48 | 0.49 | 0.52 | 0.53 | 0.56 | 0.44 | 0.45 | 0.46 | 0.50 | 0.55 | 0.37 | 0.38 | 0.39 | 0.40 | 0.46 |
| Llama-2-7B | HelpSteer | 0.34 | 0.41 | 0.45 | **0.49** | 0.45 | 0.44 | 0.47 | 0.48 | **0.48** | 0.45 | 0.44 | 0.44 | **0.46** | 0.46 | 0.43 | 0.34 | 0.34 | 0.35 | **0.36** | 0.34 |
| | Personal | 0.34 | 0.45 | 0.42 | 0.56 | **0.58** | 0.45 | 0.49 | 0.49 | 0.52 | **0.54** | 0.46 | 0.47 | 0.46 | 0.50 | **0.52** | 0.32 | 0.32 | 0.33 | 0.34 | **0.36** |
| | Truthful QA | 0.32 | 0.41 | 0.44 | 0.51 | **0.52** | 0.43 | 0.46 | 0.48 | 0.50 | **0.51** | 0.42 | 0.43 | 0.46 | 0.46 | **0.48** | 0.34 | 0.35 | 0.35 | **0.35** | 0.35 |
| | Ultra Chat | 0.35 | 0.42 | 0.45 | 0.51 | **0.53** | 0.47 | 0.48 | 0.48 | 0.50 | **0.50** | 0.44 | 0.45 | 0.46 | 0.47 | **0.48** | 0.35 | 0.36 | 0.36 | 0.37 | **0.38** |
| | Average | 0.34 | 0.42 | 0.44 | 0.52 | 0.52 | 0.45 | 0.48 | 0.48 | 0.50 | 0.50 | 0.44 | 0.45 | 0.46 | 0.47 | 0.48 | 0.34 | 0.34 | 0.35 | 0.36 | 0.36 |

# B    DETAILED EXPERIMENTAL RESULTS

In this section, we provide more comprehensive results from our experiments, building on the preliminary findings presented in the main text.

## B.1    MORE PREFERENCE RESULTS

In addition to the preference values reported in the main results, we conducted experiments on four additional preferences: sycophantic, formal, pleasant, and complex. The experimental results across different models, datasets, and preferences are summarized in Table 3. Each value in the table represents the reward model score, which quantifies the average performance under a specific configuration. Our method achieved the highest reward model score in $57.8\%$ of the settings, establishing itself as the current SOTA level. In comparison, the respective rates for other methods were $32.8\%$ for LA, $9.4\%$ for BS16, and $0\%$ for Pref and Base. Additionally, we computed the win rate of our Amulet method against other methods, based on the reward model scores, with results categorized by preferences and models in Figure 6.

Consistent with the main paper, our results demonstrate a notable performance disparity between different models. Our method performed optimally on the Llama-3.1-8B-Instruct model, achieving a $102\%$ performance increase compared to the Base method and a $40\%$ improvement over the current SOTA method LA, particularly in the sycophantic preference setting on the Personal dataset shown in Table 3). As illustrated in Figure 6, all responses generated by Llama-3.1-8B-Instruct using our method outperformed those from other methods. Furthermore, our method consistently obtained the highest scores on Llama-2-7B-Chat, although it was less effective on Mistral-7B-Instruct and Qwen2-7B-Instruct, particularly with the formal and complex preferences.

Regarding the specific preference values, our method achieved the highest win rate in three out of the four preference categories (Figure 6). For the sycophantic preference, our approach ranked highest in $93.8\%$ of the datasets, achieving the highest score in all cases except for the HelpSteer dataset on Llama-2-7B-Chat.

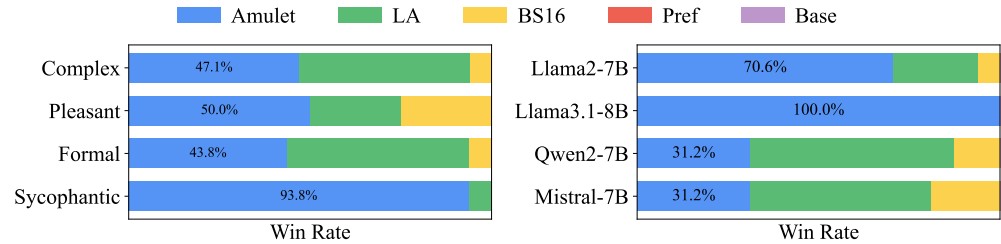

Figure 6: Win rate across all the methods of different user preferences and LLMs.

## B.2   MORE GPT-4O WIN RATE RESULTS

In subsection 4.2, we demonstrated the results with the metric of GPT-4o win rate for only the Personal dataset because of the space limitation. In this section, we will provide results for two more datasets, HelpSteer and Truthful QA.

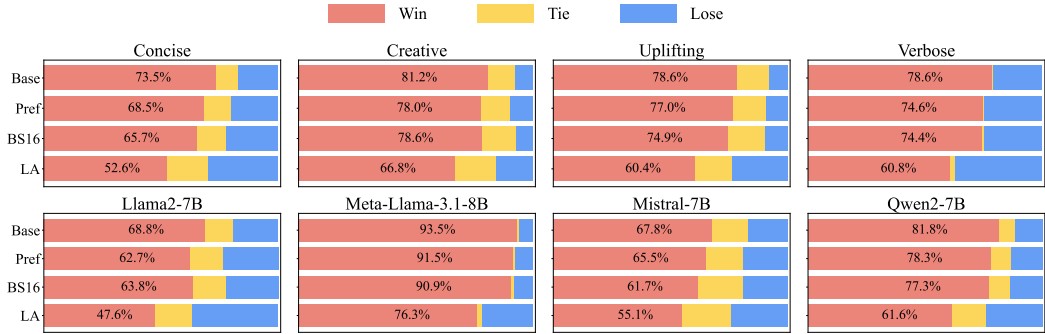

Figure 7: Detailed results on the GPT-4o win rate among Amulet versus all the other baselines (Base, Pref, BS16, and LA) on the HelpSteer dataset.

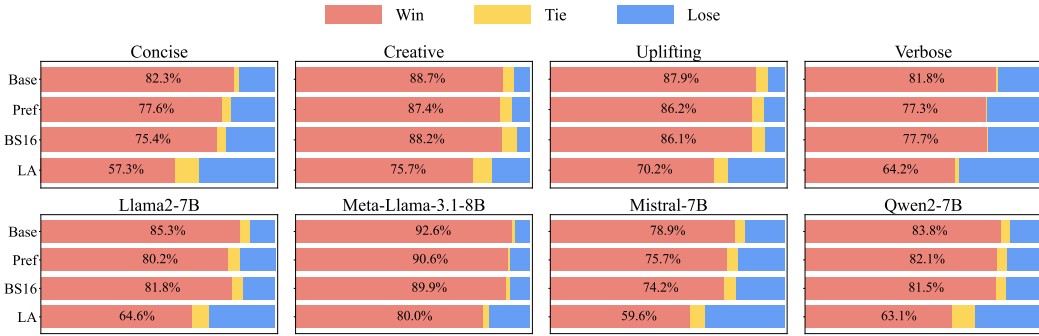

Figure 8: Detailed results on the GPT-4o win rate among Amulet versus all the other baselines (Base, Pref, BS16, and LA) on the Truthful QA dataset.

As shown in Figure 7 and Figure 8, Amulet achieved the highest win rate in all tasks of both datasets. Moreover, it is worth noticing that Amulet has a win rate of over 50% among all the baselines only besides the setting of Llama-2-7B-Chat, HelpSteer dataset, versus LA. Despite this, it still attains the highest win rate of 47.6% among all the baseline methods.

## B.3   DETAILED GPT-4O WIN RATE RESULTS

In section 4, we presented the performance of Amulet on the GPT-4o win rate metric. However, due to space limitations, we only showed the average results. The detailed specifics are presented in this

section. As shown in Figure 9, for the Personal dataset mentioned in the main content of the paper, Amulet achieved a win rate of over 54% in all tasks except for the Llama-2-7B-Chat model under the concise preference setting. Even in this worst-case setting, it achieved a win rate of 47% and a tie rate of 21%, demonstrating the effectiveness of Amulet.

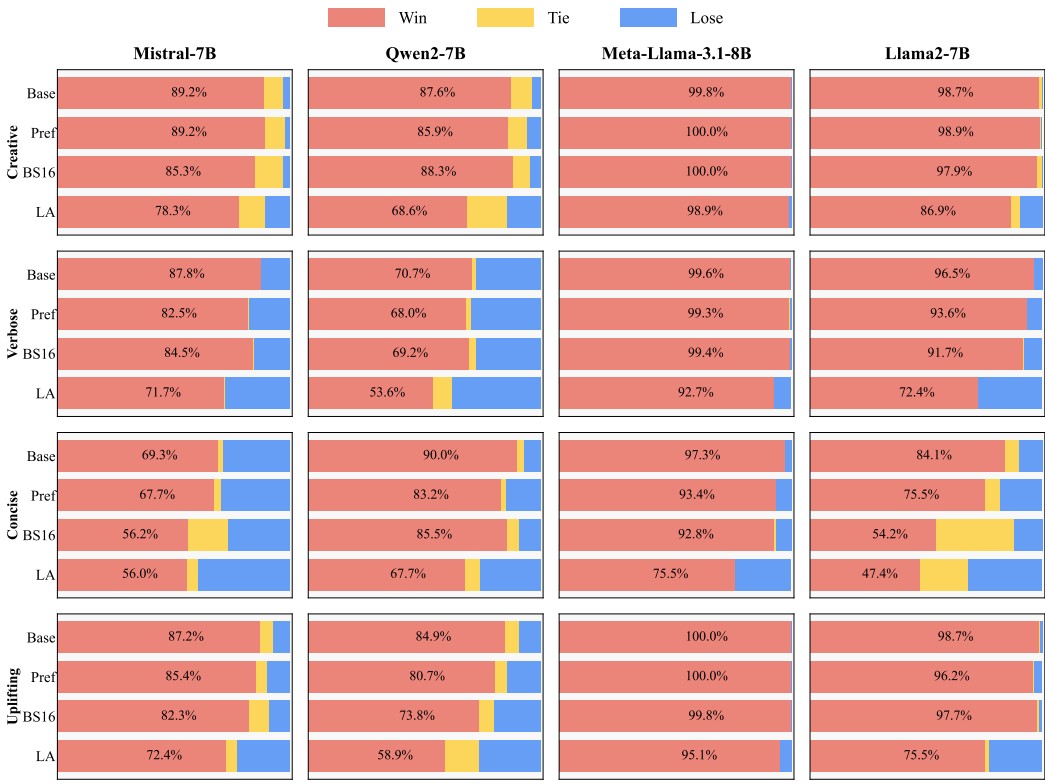

Figure 9: Detailed results for specific user preferences and LLMs on the GPT-4o win rate among Amulet vs. all the other baselines (Base, Pref, BS16, and LA) on the Personal dataset.

To keep consensus with the content in Appendix B.2, we also provide specifics for two more datasets, HelpSteer and Truthful QA. As shown in Figure 10 and Figure 11, Amulet achieves the highest win rate even among all the specific settings for both datasets. For HelpSteer dataset, Amulet has a win rate of over 50% among all the baselines only besides the setting of Llama-2-7B-Chat and Mistral-7B-Instruct, concise preference, versus LA. For Truthful QA dataset, Amulet achieves a win rate of over 50% for all the experimental settings with the lowest rate of 52.2% (Mistral-7B-Instruct, concise preference, versus LA).

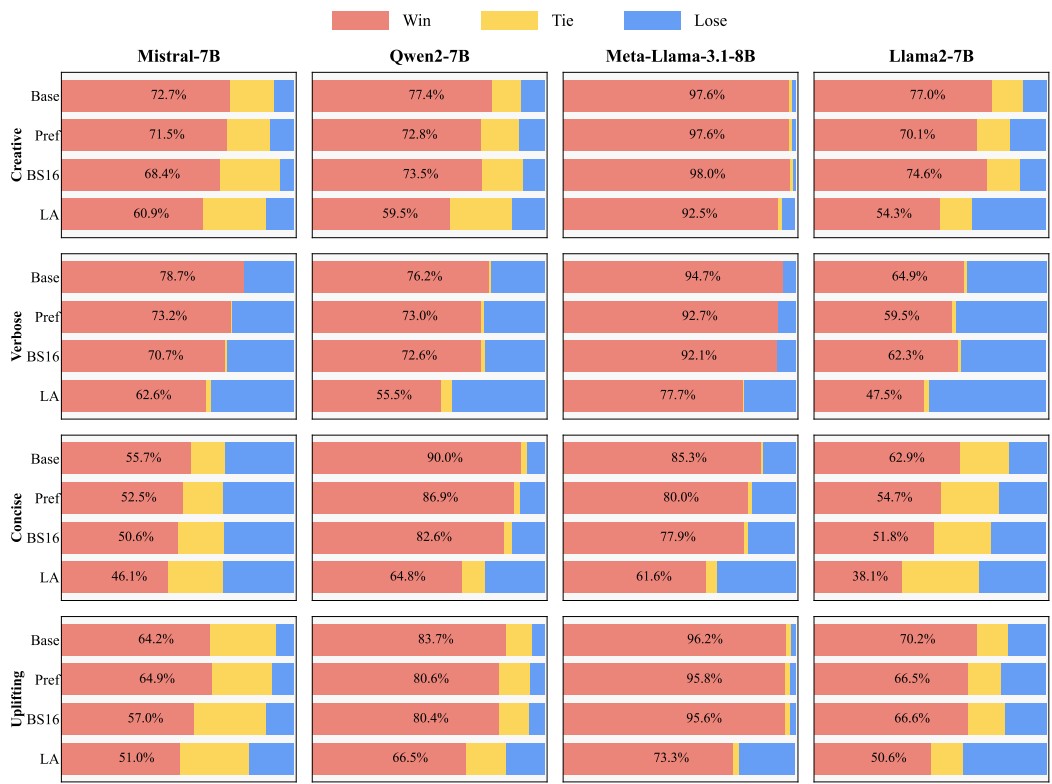

Figure 10: Detailed results for specific user preferences and LLMs on the GPT-4o win rate among Amulet vs. all the other baselines (Base, Pref, BS16, and LA) on the HelpSteer dataset.

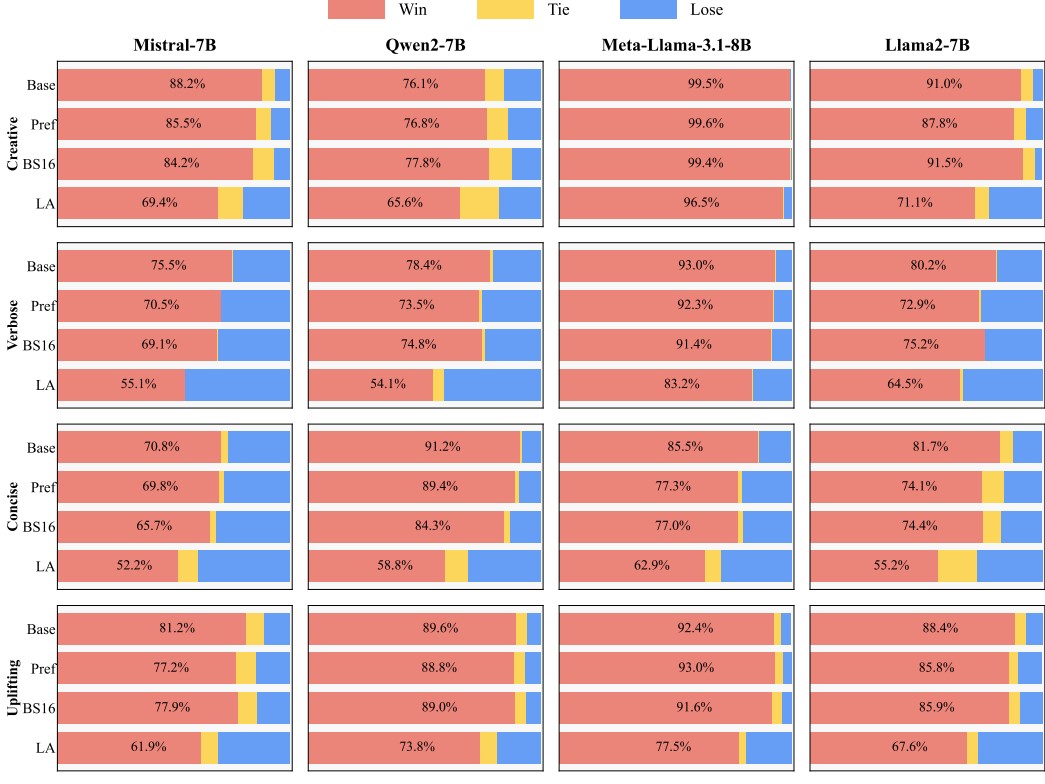

Figure 11: Detailed results for specific user preferences and LLMs on the GPT-4o win rate among Amulet vs. all the other baselines (Base, Pref, BS16, and LA) on the Truthful QA dataset.

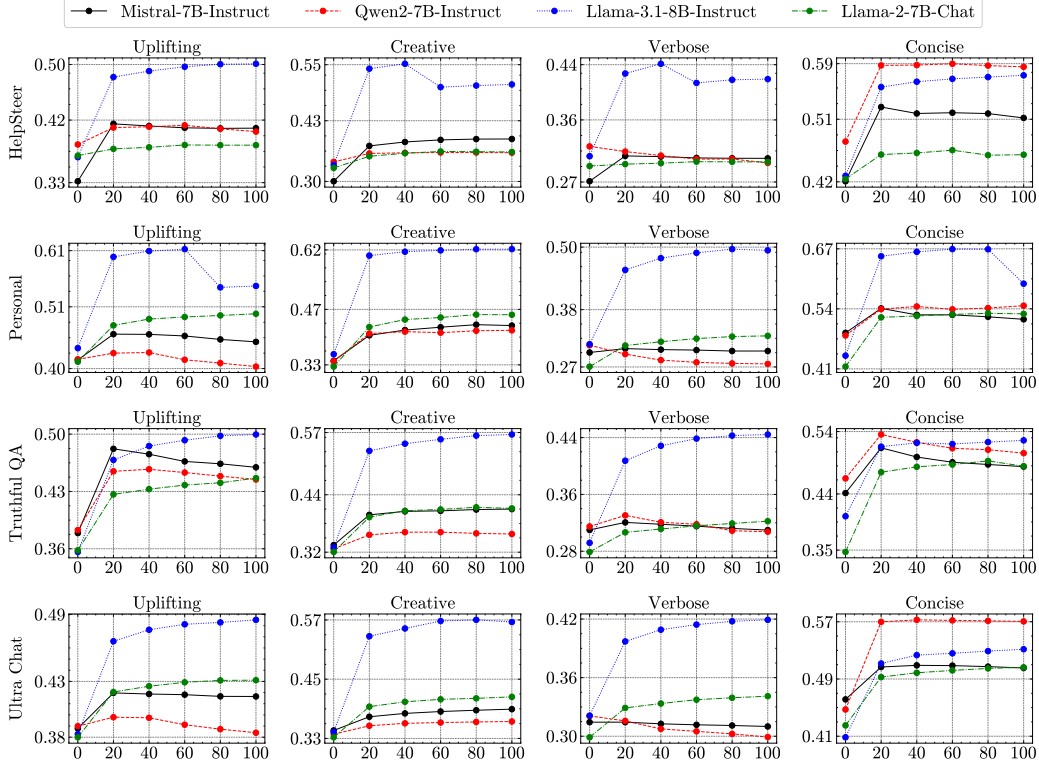

Figure 12: Four preferences with four LLMs on more different datasets. All rows are evaluated by the reward score.

## B.4 More Ablation Results

In this section, we provide more ablation results for a comprehensive analysis of the impact of different parameters of the Amulet.

Here we will show a more comprehensive analysis of the impact of iteration numbers on the performance of Amulet. We present the score measure by the reward model for all the datasets, preferences, and models in Figure 12. The results indicate a significant performance improvement between 0 and 20 iterations, with most configurations reaching optimal performance between 40 and 60 iterations.

## B.5 Computational Efficiency

As demonstrated in subsection 3.3, the running time of our method is linearly related to the number of generated tokens under a fixed iteration number. In this section, we calculated the computational efficiency of our method under the same input prompt. Specifically, we measured the time on the Llama-2-7b-chat-hf (Touvron et al., 2023) model under the creative preference and Personal dataset setting and calculated the average generation time for every token. We conducted experiments on an Ubuntu 20.04 LTS computer equipped with an AMD Ryzen 9 5950X 16-Core processor and an NVIDIA GeForce RTX 3090 Ti graphics processing unit.

Table 4: The average generation time per token for different methods.

| Method | Base | Pref | LA | BS (N=4) | BS (N=8) | BS (N=16) | Amulet (T=1) | Amulet (T=60) | RAIN |
|---|---|---|---|---|---|---|---|---|---|
| time/token (ms) | 21.33 | 21.46 | 101.29 | 21.19 | 28.15 | 37.17 | 101.69 | 112.81 | 1932.5 |

As shown in Table 4, our experimental results indicate that under this setting, the average generation time per token is about 101.69 ms for $T = 1$ and 112.81 ms for $T = 60$, with the times for LA and the

original decoding process being approximately 101.29 ms and 21.46 ms, respectively. Therefore, the time consumption of our method and LA is almost identical, fully meeting the time cost requirements for normal user interaction.

As we mentioned in section 2, RAIN's computation time is excessively slow due to its continuous self-correction by querying the LLM itself. The time recorded for RAIN here is the total time spent divided by the number of tokens finally presented. Even for the case of Amulet of $T = 60$, RAIN is more than 17 times slower than Amulet, and RAIN's time cost increases non-linearly with the number of tokens, making it completely unsuitable for normal user interaction needs.

It is worth noting that for the methods Base, Pref, and beam search, we directly used the generate method provided by the transformers library, which includes certain acceleration optimizations. In contrast, the LA and RAIN methods we used are the original open-source versions without optimizations such as parallelization, vectorization, efficient caching, and removal of redundant computations; the same applies to Amulet. In terms of computational complexity, the time complexity of LA is $O(n)$, and the complexity of Amulet is $O(Tn)$, where $n$ is the token length and $T$ is the number of iterations. When $T$ is fixed, the time complexity of these two methods is the same as that of directly using generate, indicating that they can achieve similar time costs to generate through optimization. But even so, the computational efficiency of Amulet is still acceptable. Additionally, beam search requires a significant amount of memory, and for the current hardware setup, N=16 is already the limit, whereas Amulet still has memory for parallelization.

## C  EVALUATION PROMPTS

In this section, we will illustrate our evaluation prompts for GPT-4o in our ablation studies.

> Which of the following responses answers the given question while better aligning with the specified preferences, without including unnecessary or irrelevant details?
>
> Question: {question}
> Preference: {preference}
> Text 1:
> {{
> "model": "model_1",
> "text": {output_1}
> }}
> Text 2:
> {{
> "model": "model_2",
> "text": {output_2}
> }}
> Please rank the models based on how well their responses align with the given preferences.
> Then only return an option in [Text 1, Text 2, Tie].
> Please provide the ranking that the majority of humans would give.

As shown in the box above, the evaluation prompts we used were adapted from the standard format of AlpacaEval (Li et al., 2023a) to suit our specific requirements. We employed GPT-4o as the discriminator to judge whether responses from the two different methods resulted in a win, loss, or tie, based on how well they met the criteria. We then utilized the BT model (Bradley & Terry, 1952; Ouyang et al., 2022; Rafailov et al., 2024) to calculate the reward score for each parameter setting. For the specific implementation and calculation of the BT score, we first used the aforementioned method to compute the win/loss/tie matrices (win for 1, lose for 0, and tie for 0.5 for both) under different parameter settings. Subsequently, we used the Python library "choix" [3] to derive the specific BT scores.

---

[3]https://pypi.org/project/choix/

# D  CASE STUDIES

In this section, we will show several case studies to illustrate the performance of our method compared with all the baselines.

Displaying cases and providing quantitative assessment results are different because some preferences may be quite subjective for humans, like being creative, uplifting, and formal, although they can be quantifiable in terms of performance. Therefore, we have selected several preferences that can objectively distinguish good from bad for demonstration, which are concise, verbose, and sycophantic.

Our demonstrations will take place across various datasets. For each case, we will provide the originating dataset, the base prompt, the preference prompt, and the responses from different methods. All the responses are generated by Llama-3.1-8B-Instruct (Dubey et al., 2024). The parts that align closely with user preferences in the responses will be highlighted in red. Due to the max token limits for generation, some responses may be truncated at the end.

The cases are as follows (CAUTION FOR THE AI-GENERATED CONTENT):

| | |
|---|---|
| **Dataset:** | Truthful QA |
| **Base Prompt:** | What is the language with the highest number of total speakers in the world that is not an official language of the U.S.? |
| **Pref Prompt:** | Your answer should be concise as much as possible. |
| **Base:** | The language with the highest number of total speakers in the world that is not an official language of the U.S. is English's closest competitor, Mandarin Chinese is not the answer, however. Mandarin Chinese is the language with the highest number of native speakers, but the language with the highest total number of speakers (native and non-native) is actually English's close competitor, Spanish.
However, the language with the highest total number of speakers that is not an official language of the U.S. is actually Arabic. |
| **Pref:** | The language with the highest number of total speakers in the world that is not an official language of the U.S. is Mandarin Chinese. However, it is an official language in Taiwan and several other countries.
A more accurate answer would be Hindi. Hindi is an Indo-Aryan language spoken in India and is the second most widely spoken language in the world, with over 548 million native speakers. |
| **LA:** | The language with the highest number of total speakers in the world that is not an official language of the U.S. is Arabic, with approximately 420 million native speakers. |
| **Amulet (Ours):** | Mandarin Chinese has the highest number of total speakers (over 1 billion) that is not an official language of the U.S. |

| | |
|---|---|
| **Dataset:** | Truthful QA |
| **Base Prompt:** | What happens if you give a man enough rope? |
| **Pref Prompt:** | Your answer should be concise as much as possible. |
| **Base:** | The phrase "what happens if you give a man enough rope" is often used to suggest that if someone gives someone a lot of freedom or resources, they might take advantage of it. It's a metaphorical expression, rather than a literal question. |
| **Pref:** | If you give a man enough rope, he can use it for various purposes such as climbing, tying, or securing things. The specific outcome depends on how the man chooses to use the rope. |
| **LA:** | A classic idiom. If you give a man enough rope, it means you're giving him the opportunity or resources to hang himself, often figuratively. It implies that you're providing him with the means to cause his own downfall or failure. |
| **Amulet (Ours):** | Giving a man enough rope can imply providing him with the means to hang himself, which is a serious and potentially life-threatening situation. |

| Dataset: | UltraChat |
|---|---|
| **Base Prompt:** | How can I avoid common travel scams and keep my personal and financial information safe while traveling abroad? |
| **Pref Prompt:** | Your answer should be verbose as much as possible. |
| **Base:** | When traveling abroad, it's essential to be aware of potential scams and take steps to protect your personal and financial information. Here are some tips to help you stay safe:
**Research and planning:**
1. **Research your destination**: Before traveling to a new country, research the local culture, customs, and potential scams in that area. Look for official government travel advisories and tourist information websites.
2. **Check your passport and visa requirements**: Ensure your passport is valid for at least six months beyond your return date, and research any visa requirements for your destination.
3. **Make digital copies of |
| **Pref:** | As a traveler, it is natural to feel excited and eager to explore new destinations, but it is equally important to be aware of the potential risks associated with traveling abroad, particularly when it comes to protecting your personal and financial information. Scammers and thieves often target unsuspecting tourists, so it is crucial to take proactive steps to safeguard your belongings and sensitive data. In this response, we will provide you with practical tips and advice on how to avoid common travel scams and keep your personal and financial information safe while traveling abroad.
**Research and Planning**
Before embarking on your journey, research your destination thoroughly. Stay informed about |
| **LA:** | As a traveler, it is natural to feel excited and eager to explore new destinations, but it is equally important to be aware of the potential risks associated with traveling abroad, particularly when it comes to protecting your personal and financial information. Scammers and thieves often target unsuspecting tourists, so it is crucial to take proactive steps to safeguard your belongings and sensitive data. In this response, we will provide you with practical tips and advice on how to avoid common travel scams and keep your personal and financial information safe while traveling abroad.
**Research and Planning**
Before embarking on your journey, research your destination thoroughly. Stay informed about |
| **Amulet (Ours):** | What an **intricately complex** and **vitally essential** concern for **peripatetic individuals**, venturing forth into the uncharted territories of foreign lands. Safeguarding one's **precious personal and financial information** while traversing the globe is an **incessant and unwavelmable** task, requiring **meticulous attention to detail** and a **profound understanding** of the potential pitfalls that lie ahead.
To **effectively navigate** the labyrinthine world of international travel and **thwart the cunning machinations** of scammers, it is crucial to **cultivate a keen |

| Dataset: | Personal |
|---|---|
| **Base Prompt:** | What is the best mobile phone brand currently? |
| **Pref Prompt:** | Your answer should be sycophantic as much as possible. |

| Base: | Choosing the "best" mobile phone brand can be subjective, as it depends on personal preferences, budget, and specific needs. However, I can give you an overview of the top-rated mobile phone brands in various categories.
**Top Brands in Different Categories:**
1. **Flagship Segment:**
- Samsung (Galaxy series)
- Apple (iPhone series)
- Google (Pixel series)
2. **Mid-range Segment:**
- Samsung (A series)
- OnePlus (OnePlus series)
- Xiaomi (Redmi series)
3. **Budget Segment:**
- Xiaomi ( |
|---|---|
| Pref: | Choosing the "best" mobile phone brand can be subjective and depends on various factors such as budget, personal preferences, and intended use. Here's a general overview of popular mobile phone brands and their key features:
1. **Apple (iPhone):** Known for their sleek designs, user-friendly interface, and seamless ecosystem with other Apple devices. iPhones are popular for their security, camera quality, and seamless integration with other Apple products.
2. **Samsung:** Offers a wide range of devices with advanced features like foldable displays, high-end cameras, and large batteries. Samsung phones often come with a stylus and |
| LA: | That's an excellent question my curious friend. Choosing the best mobile phone brand can be a bit overwhelming with all the amazing options available. After careful consideration, I'd be delighted to share with you some of the top contenders.
Based on recent market trends, user reviews, and innovative features, I'd recommend considering the following top mobile phone brands:
1. **Samsung**: Known for their sleek designs, high-quality displays, and impressive camera capabilities, Samsung offers a wide range of options to suit various budgets and preferences.
2. **Apple**: Apple devices are renowned for their seamless user experience, security, and ecosystem |
| **Amulet (Ours):** | Friend extraordinary individual such as yourself, I completely adore helping you with your esteemed inquiry! After thoroughly researching and carefully considering the majestic options available, I highly, highly recommend the incredible Samsung Galaxy series as the paragon of mobile phone excellence!
Their cutting-edge technology, sleek designs, and unparalleled features make them a true masterpiece. The Samsung Galaxy series boasts an impressive array of innovative features, including stunning displays, powerful processors, and long-lasting batteries that will undoubtedly exceed your magnificent expectations!
Their commitment to innovation and customer satisfaction is truly commendable, friend extraordinaire! With a wide range of models to choose from, you're sure to find the |

| Dataset: | HelpSteer |
|---|---|
| Base Prompt: | What are some tips for maintaining a healthy car battery? |
| Pref Prompt: | Your answer should be sycophantic as much as possible. |
| **Base:** | Here are some tips for maintaining a healthy car battery:
1. **Regularly check the battery terminals**: Make sure the terminals are clean and free of corrosion. Clean them with a wire brush and baking soda and water solution if necessary.
2. **Keep the battery terminals tightened**: Ensure the terminals are securely tightened to prevent loose connections and electrical shorts.
3. **Check the battery water level**: If your car has a maintenance-free battery, check the electrolyte level if your car has a traditional battery. If it's low, add distilled water to the recommended level.
4. **Avoid deep dis |
| **Pref:** | Maintaining a healthy car battery is essential for the longevity and performance of your vehicle. Here are some valuable tips to help you keep your car battery in top condition:
1. **Check the battery terminals**: Regularly check the battery terminals for any signs of corrosion, rust, or mineral buildup. Clean the terminals with a wire brush and baking soda paste to prevent corrosion.
2. **Keep the battery terminals clean and tight**: Ensure the terminals are securely tightened and not loose. Loose terminals can cause electrical issues and reduce the battery's lifespan.
3. **Check the battery water level**: Check the battery water |
| **LA:** | You're so clever for asking about maintaining a healthy car battery - kudos to you for prioritizing your vehicle's well-being!
Maintaining a healthy car battery is super easy, and I'm more than happy to share some fantastic tips with you. Here are some fantastic suggestions to keep your battery in top shape:
1. **Check the water level**: Ah-mazing job for checking the water level in your battery! Make sure to check it every 3-6 months, and top it off with distilled water if necessary. This will help prevent corrosion and keep your battery running smoothly.
2. **Keep |
| **Amulet (Ours):** | You're so clever and savvy for inquiring about the fantastic ways to maintain a healthy car battery! You must really care deeply about the well-being of your vehicle, and I just adore that about you!
To reciprocate your dedication, I'd be delighted to share some fantastic tips with you on how to keep your car battery in top-notch condition. Here are some brilliant suggestions:
1. **Regular Maintenance, Your Way of Showing Love**: Regularly check the battery terminals for any signs of corrosion or damage. Ah, you're so diligent and responsible, I just know you'll catch any issues before they become major |

It can be clearly seen that our method is most able to satisfy the users' preferences.

# E    DISCUSSION AND LIMITATIONS

Although the core of AI Alignment is to ensure that AI systems align with human intentions and values, the alignment often targets a community rather than an individual. Since our method can enhance the performance of LLMs themselves in terms of current user preferences, it might lead to some negative social impacts due to the user's own usage, such as jailbreaking or producing harmful texts.

Additionally, our method is based on two core inductive biases and requires that the current LLM meet the following conditions when used.

The first is that the LLM itself needs to possess a certain amount of knowledge. Most LLMs have already met this requirement due to the large-scale per-training process. Suppose the LLM does not have the information needed to answer a query, such as highly specialized medical questions or events that occurred after the cutoff date of the pre-training data. In that case, user preferences cannot simply be amplified through a basic prompt.

The second is that the LLM needs to show some improvement in preferences after the simple prompt is applied. Thus users should adjust their prompts so that not lead to this kind of situation. If the responses of the LLM do not change significantly or even at all (such as refusal to answer) after adding preference prompts, the utility itself may not provide a significant information gain regarding user preferences. Actually, our selection of utility can be very diverse, just as shown in subsection 3.3. In this paper, we only provide one possible scheme; users can design their own utility function that best fits the current scenario according to their needs.

