# OpenReview forum: "Amulet: ReAlignment During Test Time for Personalized Preference Adaptation of LLMs"
_ICLR.cc/2025/Conference — ICLR 2025 Poster_

### Official Review · Reviewer_dMQ4 · 2024-11-01

**Soundness:** 4
**Presentation:** 3
**Contribution:** 3
**Rating:** 6
**Confidence:** 4

**Summary:**

This paper proposes a new framework, Amulet, for single-user preference alignment at test time. Amulet treats each token generation step as an independent online learning problem and achieves real-time, personalized preference alignment without requiring model retraining or fine-tuning by introducing a closed-form solution. Experimental results demonstrate Amulet’s strong alignment performance and efficiency across various combinations of LLMs, datasets, and preference dimensions.

**Strengths:**

Amulet offers an innovative test-time preference alignment method, enabling real-time adaptation to user preferences during generation without fine-tuning and thereby reducing resource costs.

The closed-form solution design significantly enhances computational efficiency, where detailed theoretical analysis is provided.

Validation across different LLM and dataset combinations shows Amulet’s robustness and broad applicability in multiple scenarios.

**Weaknesses:**

The paper only compares with Pref and Linear Alignment (LA), lacking comparisons with other mainstream alignment methods. To improve credibility, it’s recommended to introduce more alignment methods as baselines on at least one foundation model.

Amulet assumes sufficient user history for preference alignment, but its performance with insufficient user history is not discussed, impacting the method’s practicality.

**Questions:**

How does Amulet perform if user history information is insufficient or lacking?

How does Amulet perform while compared with more SOTA baselines?

---

> ### Author Response · Authors · 2024-11-20
>
> Thank you, reviewer dMQ4, for your constructive feedback. Here are our responses, organized to address each of your concerns one by one specifically:
>
> **Q1.The paper only compares with Pref and Linear Alignment (LA), lacking comparisons with other mainstream alignment methods. To improve credibility, it’s recommended to introduce more alignment methods as baselines on at least one foundation model.**
> > **A1**: Thank you for your question for the understanding that **there is indeed a lack of relevant baselines in the current field**. We have additionally added four baselines: three different versions of beam search (BS) and RAIN [1]. We used the Llama-2-7b-chat model and conducted experiments on the Personal and Truthful QA datasets, as well as on the creative and verbose preferences. The results are shown in the table below:
>
> **Results of more baselines on Personal dataset**
> |                  |   Base |   Pref |   LA |   Amulet | BS (N=4) | BS (N=8) | BS (N=16) |
> |:-----------------|------------:|------------:|-------------:|-------------:|-------------:|-------------:|-------------:|
> | creative   |     0.32 |     0.33 |     0.39   |     **0.45**   |  0.33 | 0.32 | 0.32 |
> | verbose  |     0.26 |     0.27 |      0.29 |     **0.32**   |     0.27   |  0.27 | 0.27 |
>
> **Results of more baselines on Truthful QA dataset**
> |                  |   Base |   Pref |   LA |   Amulet | BS (N=4) | BS (N=8) | BS (N=16) |
> |:-----------------|------------:|------------:|-------------:|-------------:|-------------:|-------------:|-------------:|
> | creative   |     0.30 |     0.32 |     0.36   |     **0.41**   |  0.31 | 0.31 | 0.31 |
> | verbose  |     0.27 |     0.28 |      0.30 |     **0.32**   |     0.28   |  0.28 | 0.28 |
>
> > As the results illustrated, even with the addition of four extra baselines, Amulet still achieved the best performance across all tasks. It is worth noting, as mentioned in the article, that RAIN's running efficiency is very low (more than 17 times slower than Amulet, more details are provided in A2 of the reviewer KgK2). Therefore, we prioritized completing the experiments of RAIN on the Personal dataset, with results of 0.39 for creative and 0.26 for verbose, whereas Amulet achieves 0.45 and 0.32 respectively. Even though RAIN took much more time than Amulet, its performance still falls far short of our method. We will complete the RAIN experiments of Truthful QA dataset and add this experiment to the main experiments part of the revision version.

---

> > ### Comment · Reviewer_dMQ4 · 2024-11-20
> >
> > Thanks for your effort! The response addresses my concern Q1.

---

> ### Author Response · Authors · 2024-11-20
>
> **Q2.Amulet assumes sufficient user history for preference alignment, but its performance with insufficient user history is not discussed, impacting the method’s practicality.**
> > **A2**: Thank you for your valuable question. In this paper, we investigated how to achieve real-time preference adaptation for LLMs without any fine-tuning, just as illustrated in our Figure 1. Hence, we do not need sufficient user history; we only need to address the specific preferences indicated by the user during the current interaction / time.
> >
> > Therefore, what we should actually discuss is how Amulet can be applied in scenarios with more user history. To address your concerns, we discuss two potentially applicable approaches when user history is sufficient:
> > 1. The user engages in multiple rounds of interaction with the LLM, using user history to continuously clarify the user's more specific needs, as described in [2] and [3], and then determining a specific user goal. Finally, achieving better alignment through Amulet based on the clarified goal.
> > 2. Also the scenario of multiple rounds of interaction with the user. However, we utilize Amulet to align with the user's preferences in each round of interaction, and then use all historical interaction information as a prompt to assist the next round of generation, thereby achieving better alignment.
> >
> > We will make the part your concern more clear in the discussion part of the revision version.
>
>
> **Reference:**
>
> [1] Li Y, Wei F, Zhao J, et al. RAIN: Your Language Models Can Align Themselves without Finetuning. ICLR24. (https://arxiv.org/abs/2309.07124)
>
> [2] Mechergui M, Sreedharan S. Goal Alignment: Re-analyzing Value Alignment Problems Using Human-Aware AI. AAAI24. (https://ojs.aaai.org/index.php/AAAI/article/view/28875)
>
> [3] Ma R, Qu J, Bobu A, et al. Goal Inference from Open-Ended Dialog. (https://arxiv.org/abs/2410.13957)

---

> > ### Comment · Reviewer_dMQ4 · 2024-11-20
> >
> > The response to Q2 does not fully address my concern.
> >
> > One of the key advantages of the proposed framework is its efficiency without fine-tuning. However, it requires user history for alignment. I am curious about its performance when historical data is limited, similar to cold-start scenarios in recommender systems.
> >
> > I would appreciate results under a setting with insufficient user data, as I believe the history length significantly impacts the alignment process.
> >
> > Additionally, could you clarify the statement: *"we do not need sufficient user history; we only need to address the specific preferences indicated by the user during the current interaction/time"*? Based on my experience:
> > 1. Current interaction alone is often insufficient to capture accurate preferences; historical data is typically necessary.
> > 2. A long history can introduce noise, potentially degrading model performance.
> > 3. Limited data may be inadequate to extract meaningful individual preferences.
> >
> > There might be some misunderstanding on my part, and I would appreciate it if the authors could help clarify. Thank you.

---

> > > ### Author Response · Authors · 2024-11-20
> > >
> > > Thank you for your feedback and explaining your concerns. We would like to clarify that in our setting, users directly provide explicit preferences in the form of prompts that reflect their current needs.
> > >
> > > Unlike cold-start scenarios in recommender systems, we have already known the specific user preference, and the problem is how to adapt LLM to the preference in a lightweight, real-time, and training-free way with better performance. Amulet solves the problem by optimizing the decoding process of every token with a FTRL-like closed-form solution to make more sufficient use of the user-provided prompts to provide guidance.
> > >
> > > Therefore, as shown in A2, Amulet does not need long user history, but only one-turn interaction.
> > >
> > > We hope this explanation resolves any confusion regarding the scope and purpose of our method. If further clarification is needed, we are happy to provide additional details.

---

> > > > ### Comment · Reviewer_dMQ4 · 2024-11-20
> > > >
> > > > Thank you for your explanation. I consider this work to be above the accepted threshold.

---

> > > > > ### Author Response · Authors · 2024-11-20
> > > > >
> > > > > Thank you for your acknowledgment!

---

### Official Review · Reviewer_KgK2 · 2024-11-04

**Soundness:** 3
**Presentation:** 2
**Contribution:** 3
**Rating:** 6
**Confidence:** 3

**Summary:**

This paper investagates the issue that current LLM alignment methods focus on static datasets rather than personalized, changing user preferences. Besides, the authors recognize that retraining models for each new preference is costly and impractical. Therefore, the authors introduce a training-free framework that allows real-time optimization of LLM outputs to match user preferences during test time, namely Amulet, which formulates each token generation as a separate online learning problem guided by user prompts. Experiments demonstrate notable performance improvements across different LLMs, datasets and preference types.

**Strengths:**

The paper is well-organized. It is reasonable for me to perform test time alignment in LLMs for light-weight preference optimization.

The proposed online learning decoding process by formulating each token generation as a separate online learning problem seems novel. Besides, the authors provide a closed-form solution to reduce computational costs

Experiments with several datasets and backbone LLMs demonstrate the effectiveness of the proposed method.

**Weaknesses:**

Insufficient baselines. Only LA is used as the baseline model, likely due to the scarcity of tuning-free test time alignment approaches. However, other baselines from related topics, such as those introduced in the related work, could be adapted to verify the effectiveness.

One major merit of tuning-free test time alignment is light-weighted. To this end, the time and computational complexity could be analyzed. The computational cost could be reported.

Broken sentences, such as Line 209: 'Since we provide a general framework that is unrelated to the utility.'

**Questions:**

See the above Strengths and Weaknesses.

---

> ### Author Response · Authors · 2024-11-20
>
> Thank you, reviewer KgK2, for your constructive feedback. Here are our responses, organized to address each of your concerns one by one specifically:
>
> **Q1.Insufficient baselines.**
> > **A1**: Thank you for your question and for the understanding that **there is indeed a lack of relevant baselines in the current field**. We have additionally added four baselines: three different versions of beam search (BS) and RAIN [1]. We used the Llama-2-7b-chat model and conducted experiments on the Personal and Truthful QA datasets, as well as on the creative and verbose preferences. The results are shown in the table below:
>
> **Results of more baselines on Personal dataset**
> |                  |   Base |   Pref |   LA |   Amulet | BS (N=4) | BS (N=8) | BS (N=16) |
> |:-----------------|------------:|------------:|-------------:|-------------:|-------------:|-------------:|-------------:|
> | creative   |     0.32 |     0.33 |     0.39   |     **0.45**   |  0.33 | 0.32 | 0.32 |
> | verbose  |     0.26 |     0.27 |      0.29 |     **0.32**   |     0.27   |  0.27 | 0.27 |
>
> **Results of more baselines on Truthful QA dataset**
> |                  |   Base |   Pref |   LA |   Amulet | BS (N=4) | BS (N=8) | BS (N=16) |
> |:-----------------|------------:|------------:|-------------:|-------------:|-------------:|-------------:|-------------:|
> | creative   |     0.30 |     0.32 |     0.36   |     **0.41**   |  0.31 | 0.31 | 0.31 |
> | verbose  |     0.27 |     0.28 |      0.30 |     **0.32**   |     0.28   |  0.28 | 0.28 |
>
> > As the results illustrated, even with the addition of four extra baselines, Amulet still achieved the best performance across all tasks. It is worth noting, as mentioned in the article, that RAIN's running efficiency is very low (more than 17 times slower than Amulet, more details are provided in A2). Therefore, we prioritized completing the experiments of RAIN on the Personal dataset, with results of 0.39 for creative and 0.26 for verbose, whereas Amulet achieves 0.45 and 0.32 respectively. Even though RAIN took much more time than Amulet, its performance still falls far short of our method. In Appendix B.3 of our revised manuscript, we have included additional baselines, such as beam search with beam numbers 4, 8, and 16, as well as RAIN [1].

---

> ### Author Response · Authors · 2024-11-20
>
> **Q2.One major merit of tuning-free test time alignment is light-weighted. To this end, the time and computational complexity could be analyzed. The computational cost could be reported.**
> > **A2**: Thank you for your question. The computational efficiency of the algorithm is indeed very important. We have already demonstrated some results of this in Appendix B.3. To further address your concern, we have added more experiments, as shown in the table below:
>
> |   Method   |   time / token (ms)|
> |:-----|------------:|
> | Base |       21.33 |
> | Pref |       21.46 |
> | LA   |       101.29 |
> | BS (N=4)   |       21.19 |
> | BS (N=8)   |       28.15 |
> | BS (N=16)   |       37.17 |
> | Amulet (T=1)   |       101.69 |
> | Amulet (T=60)   |       112.81 |
> | RAIN  |       1932.5 |
>
> > The same as those described in Appendix B.3 regarding the hardware settings, we conducted all these experiments on an Ubuntu 20.04 LTS PC equipped with an AMD Ryzen 9 5950X 16-Core processor and an NVIDIA GeForce RTX 3090 Ti graphics processing unit. We measured the time on the Llama-2-7b-chat-hf model under the creative preference and Personal dataset setting, and calculated the average time.
> >
> > Our experimental results indicate that under this setting, the average generation time per token is about 101.69 ms for $T$ = 1 and 112.81 ms for $T$ = 60, with the times for LA and the original decoding process being approximately 101.29 ms and 21.46 ms, respectively. Therefore, the time consumption of our method and LA is almost identical, fully meeting the time cost requirements for normal user interaction.
> >
> > As we mentioned in the paper, RAIN's computation time is excessively slow due to its continuous self-correction by querying the LLM itself. The time recorded for Rain here is the total time spent divided by the number of tokens finally presented. Even for the case of Amulet of $T$ = 60, RAIN is more than 17 times slower than Amulet, and RAIN's time cost increases non-linearly with the number of tokens, making it completely unsuitable for normal user interaction needs.
> >
> > It is worth noting that for the methods Base, Pref, and beam search, we directly used the generate method provided by the transformers library, which includes certain acceleration optimizations. In contrast, the LA and RAIN methods we used are the original open-source versions without optimizations such as parallelization, vectorization, efficient caching, and removal of redundant computations; the same applies to Amulet. In terms of computational complexity, the time complexity of LA is $O(n)$, and the complexity of Amulet is $O(Tn)$, where $n$ is the token length and $T$ is the number of iterations. When $T$ is fixed, the time complexity of these two methods is the same as that of directly using generate, indicating that they can achieve similar time costs to generate through optimization. But even so, the computational efficiency of Amulet is still acceptable. Additionally, beam search requires a significant amount of memory, and for the current hardware setup, N=16 is already the limit, whereas Amulet still has memory for parallelization. We have provided a brief introduction in lines 270–272 and included Appendix B.5 for a detailed analysis of the computational efficiency of Amulet compared to other baseline algorithms in our revised manuscript.
>
>
> **Q3.Broken sentences, such as Line 209: 'Since we provide a general framework that is unrelated to the utility.'**
> > **A3**: Thank you for pointing out this issue. We have corrected this sentence in the revised version.
>
> **Reference:**
>
> [1] Li Y, Wei F, Zhao J, et al. RAIN: Your Language Models Can Align Themselves without Finetuning. ICLR24. (https://arxiv.org/abs/2309.07124)

---

> ### Author Response · Authors · 2024-11-24
>
> Dear Reviewer KgK2,
>
> Thank you for your insightful feedback and valuable comments on our paper. We have carefully addressed the concerns you raised, as outlined in the revised version. Could you kindly confirm if the revisions adequately address the issues? If there are any remaining points that require further discussion or clarification, we would be more than willing to engage and improve the work accordingly.
>
> Additionally, if you find the changes satisfactory, we would appreciate your consideration for adjusting the score to reflect the improvements.
>
> Looking forward to your feedback.

---

> ### Author Response · Authors · 2024-12-03
> **Supplementary experiment for Q1**
>
> Thank you for raising the concern about the lack of more baselines in our experiment and for understanding that **there is indeed a lack of relevant baselines in the current field**. To further address your concern, we have supplemented additional results under all our main experimental settings (across 4 preferences, 4 datasets, and 4 LLMs) using beam search (N = 4, 8, and 16) as additional baselines. We use the reward model score as the evaluation metric, and the experimental results are shown in the anonymous link: https://anonymous.4open.science/r/amulet-detailed-experiment-results-2E46/README.md.
>
> The results show that our method, Amulet, performs best under 75% of the task setting even after introducing three more baselines. Specifically, our method achieved the best performance in creative preference, with the optimal performance in 93.75% of the settings, followed by uplifting (81.25%), and concise (75%). Even in the worst-performing verbose category, the proportion of settings where the optimal performance was achieved reached 50%. For different LLMs, our method is most effective for Llama-3.1-8B-Instruct, achieving optimal results in 100% of the settings, followed by Llama-2-7B-Chat (75%), Mistral-7B-Instruct-v0.2 (68.75%), and QWen2-7B-Instruct (56.25%). Amulet achieves the best performance in over 50% of the settings across all LLMs, demonstrating its effectiveness and versatility.

---

### Official Review · Reviewer_VESu · 2024-11-05

**Soundness:** 2
**Presentation:** 3
**Contribution:** 2
**Rating:** 5
**Confidence:** 4

**Summary:**

This paper proposes a framework called AMULET for personalization alignment at test time. It borrows ideas from Contrastive Decoding and FTRL. During inference, it obtains two token generation probability distributions based on a base prompt and a user-specific personalized prompt, treating each step of token generation as an independent online optimization problem. The authors model this optimization problem and provide a closed-form solution to balance personalization and performance.

**Strengths:**

Strengths:
1. The general idea is a hot topic in the field of LLMs (personalization alignment and test-time personalization alignment).
2. The idea of treating each step of token generation as an independent optimization problem to solve test-time alignment is an interesting exploration.
3. The paper is well-written, and the methods and formulas are easy to understand.

**Weaknesses:**

Weaknesses:
1. The experimental setup is not very clear. How did the authors tune the hyperparameters? See Questions below.
2. Based on my understanding, the method described in the paper ultimately results in a weighted of the base prompt generation probability, user-specific prompt generation probability. Therefore, the authors should provide a baseline that only adjusts the fine-tuned $\alpha$, which essentially becomes Contrastive Decoding.
3. The authors emphasize the importance of real-time preference alignment but still conduct tests on statistical datasets, which cannot fully validate their claim. I recommend the authors supplement experiments on streaming data (such as recommender systems).

**Questions:**

1. Although the reward model used by the author showed excellent performance on the benchmark, the results presented in the paper indicate that the llama series of models experienced the most significant performance improvement. I suppose this might be related to the choice of the reward model. So, are there results using 4o to evaluate Figure 2 and Figure 5?
2. Experimental setting: Is T fixed in the experiments, or is early stopping based on some metrics? What is the tuning range for other parameters, and what hyperparameters were ultimately selected? Additionally, if the authors directly tune hyperparameters on the test set, it could lead to data leakage.
3. How are the user-specific preference prompts for each dataset constructed?
4. Has the author tried other model sizes, either smaller or larger models?
5. The datasets selected by the author, aside from Personal Preference Eval, do not seem to have strong requirements for personalization. Could the author provide a more detailed explanation for the selection of the remaining three datasets, with a focus on their relevance to personalization?

---

> ### Author Response · Authors · 2024-11-20
>
> Thank you, reviewer VESu, for your constructive feedback. Here are our responses, organized to address each of your concerns one by one specifically:
>
> **Q1.The experimental setup is not very clear. How did the authors tune the hyperparameters? Is T fixed in the experiments, or is early stopping based on some metrics? What is the tuning range for other parameters, and what hyperparameters were ultimately selected? Additionally, if the authors directly tune hyperparameters on the test set, it could lead to data leakage.**
> > **A1**: Thank you for your question. In the main experiment of this paper, $T$ is fixed ($T$ = 60). We have provided an ablation study in Section 4.3 to explain why this parameter was chosen. In fact, the performance of $T$ varies between 40 and 80 across different settings, and for generality and fairness, we chose $T$ = 60.
> >
> > Regarding the adjustment range of other parameters, we also introduce this in the ablation section of the experiment (Section 4.3). Our ablation study aims to discuss the impact of parameters rather than tuning for performance. We fixed all other parameters and analyzed the impact of each parameter on the performance individually. The range for $\eta$ was 2, 4, 6, 8, $\dots$, 20, and we ultimately chose 10; the ranges for $\alpha$ and $\lambda$ were both 1, 2, 3, $\dots$, 10, and we ultimately chose 2 for both.
> >
> > Regarding tuning hyperparameters on the test set, we do not believe this will affect our experimental results for the following three reasons:
> > 1. We only tuned hyperparameters on a very small part of our experiment, specifically the "Personal dataset using Llama-3.1-8B-Instruct for the creative preference dimension" (lines 512-513), which accounts for only 1/64 of our main experiments and 1/128 of all experiments;
> > 2. The hyperparameters we selected have a certain degree of generality. Intuitively, different data, preferences, and LLMs should be sensitive to hyperparameters, but we achieved performance improvements across various tasks using only one set of hyperparameters;
> > 3. The content we presented in the ablation study was more focused on discussing the impact of parameters rather than tuning for performance. We did not ultimately choose the parameters that gave us the highest performance (more details are provided in A2).
> >
> > The above content demonstrates that our selection of hyperparameters was not tailored to our experimental environment.
> >
> > We have revised Section 4.3 to include the tuning range for each hyperparameter. We have also updated Appendix B.4 to provide additional details on the ablation experiments.
>
> **Q2.Based on my understanding, the method described in the paper ultimately results in a weighted of the base prompt generation probability, user-specific prompt generation probability. Therefore, the authors should provide a baseline that only adjusts the fine-tuned $\alpha$, which essentially becomes Contrastive Decoding.**
> > **A2**: Amulet is completely different from contrastive decoding. Although the final result is indeed some kind of fusion of the two aforementioned probabilities, the fusion method is quite complex.
> >However, we believe that the point you raised is indeed necessary. Since our ablation study is more focused on discussing the impact of parameters rather than tuning for performance, we did not adjust the value of $\alpha$ to enhance the algorithm's performance. Given that $\alpha$ and $\lambda$ influence each other to some extent, we conducted additional performance comparisons for different $\alpha$-$\lambda$ pairs and included them as an additional set of baselines. The specific results are shown in the table below:
>
> |   $\alpha$-$\lambda$   |   RM Score |
> |:-----|------------:|
> | 1.0-1.0 |       0.5835 |
> | 2.0-2.0 |       *0.5838* |
> | 3.0-3.0   |       0.5842 |
> | 4.0-4.0   |       0.5845 |
> | 5.0-5.0   |       0.5844 |
> | 6.0-6.0   |       0.5844 |
> | 7.0-7.0   |       0.5847 |
> | 8.0-8.0   |       0.5847 |
> | 9.0-9.0  |       **0.5849** |
> | 10.0-10.0  |       **0.5849** |
>
> >As shown in the table, the performance of Amulet continues to increase with the increase of $\alpha$, reaching its maximum values at 9.0 and 10.0 (bolded parts). However, the parameter we ultimately chose in the main text is 2.0 (italicized part), indicating that we did not tune the parameters solely for performance.
> >
> >For the experiment where all other parameters remain fixed and only $\alpha$ is varied, the setup has been detailed in Section 4.3 in our revised manuscript. Additional results on the performance of different Amulet variants formed by adjusting $\alpha$-$\lambda$ pairs have been provided in Appendix B.4.

---

> ### Author Response · Authors · 2024-11-20
>
> **Q3.The authors emphasize the importance of real-time preference alignment but still conduct tests on statistical datasets, which cannot fully validate their claim. I recommend the authors supplement experiments on streaming data (such as recommender systems).**
> > **A3**: Thank you for raising this question. The "real-time" mentioned in this paper refers to a kind of alignment method at test-time that does not require further fine-tuning, just as illustrated in our Figure 1. Although we used a static dataset, we only utilized the question parts of these datasets, and the preferences themselves can be specified by the user based on their needs.
> >
> > We have supplemented our "real-time" setting in Section 3.3 (lines 232-235) and emphasized its distinction from recommendation systems in our revised manuscript.
>
> **Q4.Although the reward model used by the author showed excellent performance on the benchmark, the results presented in the paper indicate that the llama series of models experienced the most significant performance improvement. I suppose this might be related to the choice of the reward model. So, are there results using 4o to evaluate Figure 2 and Figure 5?**
> > **A4**: Thank you for recognizing the performance of our method. Indeed, as you mentioned, using a single reward model as the evaluation method may introduce some bias in the results. Therefore, we supplemented our experiments on the Personal dataset by calculating the GPT-4o win rate of Amulet versus all the other baselines. Due to space limitations, we only present the average results for different models and preference dimensions (more details are presented in the anonymous link provided later), as shown in the two tables below:
>
> **Amulet vs. all the baselines on different LLMs**
> |                                   |   win_ratio |   tie_ratio |   lose_ratio |
> |:----------------------------------|------------:|------------:|-------------:|
> | Llama-2-7B-Chat        |     85.3858 |     3.46583 |     11.1467  |
> | Meta-Llama-3.1-8B-Instruct |     95.9708 |     0.075   |      3.95333 |
> | Mistral-7B-Instruct-v0.2   |     78.0667 |     4.48583 |     17.4458  |
> | Qwen2-7B-Instruct          |     74.9858 |     6.84417 |     18.1708  |
>
> **Amulet vs. all the baselines on different preference dimensions**
> |                  |   win_ratio |   tie_ratio |   lose_ratio |
> |:-----------------|------------:|------------:|-------------:|
> | concise   |     75.6075 |     4.51583 |     19.875   |
> | creative  |     90.1758 |     5.64083 |      4.18083 |
> | uplifting |     86.2508 |     3.65    |     10.0992  |
> | verbose   |     82.375  |     1.06417 |     16.5617  |
>
> > The calculation method for these two tables is to determine the GPT-4o win rate of Amulet vs. LA, Pref, and Base across different LLMs and preference dimensions, and then average the results for each LLM and preference dimension. We used the standard GPTEval prompt from AlpacaEval [2], which is also displayed in Appendix C. We also provide more detailed win rates for each baseline, which are shown in the following anonymous link: https://anonymous.4open.science/r/amulet-detailed-experiment-results-2E46/README.md
> >
> > As shown in these two tables and the anonymous link, Amulet's performance under GPT-4o is significantly better than the ArmoRM-8B reward model. In our paper, Amulet's performance in the Qwen2-7B-Instruct verbose setting is not as good as other baselines. However, when evaluated with GPT-4o, Amulet significantly surpasses all baselines in all the settings, achieving the best performance.
> >
> > In our revised manuscript, we have added the experimental results and analysis using the GPT-4o win rate metric in Section 4.2, with the findings visualized in Figure 3. Further details on the GPT-4o win rate experiments have been provided in Appendix B.2 due to space limitations in the main text.
>
> **Q5.How are the user-specific preference prompts for each dataset constructed?**
> > **A5**: As shown in Appendix D, the prompt we constructed is very simple, just: "Your answer should be {preference} as much as possible." We used this uniform user-specific preference prompt for all our datasets. There are two reasons for constructing the prompts in this way:
> > 1. This is how the preference prompt is constructed in our important baseline, Linear Alignment (LA);
> > 2. We aim to minimize the impact of the prompt itself on the algorithm. Although this prompt is very simple, as our experiments have shown, such simplicity can significantly enhance the alignment performance.

---

> ### Author Response · Authors · 2024-11-20
>
> **Q6.Has the author tried other model sizes, either smaller or larger models?**
> > **A6**: Thank you for raising this question. We believe this issue is indeed very important, so we have added experiments with two small models, Qwen2-0.5B-Instruct and llama-3.2-1B-Instruct, and two big models, llama-2-13B-Chat and llama-2-70B-Chat. All experiments were conducted on the Personal dataset.
>
> **Results of the creative preference**
> |                  |   Base |   Pref |   LA |   Amulet |
> |:-----------------|------------:|------------:|-------------:|-------------:|
> | Qwen2-0.5B-Instruct   |     0.2734 |     **0.2939** |     0.2658   |     0.2861   |
> | Llama-3.2-1B-Instruct |   0.2776   |    0.2665     |    0.3371   |     **0.3507**   |
> | Llama-2-13B-Chat   |     0.2819 |     0.3051    |     0.4206  |     **0.4562**   |
> | Llama-2-70B-Chat   |     0.3262  |     0.3327 |     0.3911  |     **0.4267**   |
>
> **Results of the uplifting preference**
> |                  |   Base |   Pref |   LA |   Amulet |
> |:-----------------|------------:|------------:|-------------:|-------------:|
> | Qwen2-0.5B-Instruct   |     0.3268 |     0.345 |     0.3625   |     **0.4065**   |
> | Llama-3.2-1B-Instruct |  0.3408    |    0.315     |   **0.3905**    |    0.3834    |
> | Llama-2-13B-Chat   |     0.3503 |     0.3718    |     0.4492  |     **0.4774**   |
> | Llama-2-70B-Chat   |     0.4015  |     0.4071 |     0.4375  |     **0.471**   |
>
> **Results of the concise preference**
> |                  |   Base |   Pref |   LA |   Amulet |
> |:-----------------|------------:|------------:|-------------:|-------------:|
> | Qwen2-0.5B-Instruct   |     0.3129 |     0.3404 |     0.3333   |     **0.3735**   |
> | Llama-3.2-1B-Instruct |   0.3128   |     0.2917    |   0.3843    |    **0.3848**    |
> | Llama-2-13B-Chat   |     0.3394 |     0.3788    |     0.4915  |     **0.5364**   |
> | Llama-2-70B-Chat   |     0.4427  |     0.5053 |     0.5665  |     **0.6223**   |
>
> **Results of the verbose preference**
> |                  |   Base |   Pref |   LA |   Amulet |
> |:-----------------|------------:|------------:|-------------:|-------------:|
> | Qwen2-0.5B-Instruct   |     0.2295 |     **0.2486** |     0.2113   |     0.226   |
> | Llama-3.2-1B-Instruct |   0.2311   |    0.2204     |    0.3282   |    **0.3638**    |
> | Llama-2-13B-Chat |     0.2397 |     0.2517    |     0.2989  |     **0.3323**   |
> | Llama-2-70B-Chat |     0.28  |     0.2831 |     0.3157  |     **0.329**   |
>
> > As the table shows, Amulet also demonstrates excellent performance across different model sizes. For Llama-2-13B-Chat and Llama-2-70B-Chat, Amulet achieved the best performance in all four preferences. For the small model Llama-3.2-1B-Instruct, Amulet achieved the best performance in all the preferences except uplifting. Even for the very small model Qwen2-0.5B-Instruct, it achieved the best results in half of the tasks.
> >
> > In our revised manuscript, we have analyzed the influence of varying model sizes on Amulet’s performance in Section 4.3, with the experimental results summarized in Table 2 for clarity.

---

> ### Author Response · Authors · 2024-11-20
>
> **Q7.The datasets selected by the author, aside from Personal Preference Eval, do not seem to have strong requirements for personalization. Could the author provide a more detailed explanation for the selection of the remaining three datasets, with a focus on their relevance to personalization?**
> > **A7**: Thank you for your question. We chose these datasets for the following three reasons:
> > 1. The metric we use, the ArmoRM-8B reward model, is trained on the Ultrafeedback datasets (including Truthful_qa and UltraChat, etc.) and the Helpsteer dataset. Therefore, using these datasets allows the reward model to provide more accurate scores.
> > 2. The Ultrafeedback and Helpsteer datasets are themselves multi-objective preference datasets, and have been widely used in the alignment field to reflect user preferences, especially in works dealing with multi-objective preference alignment, such as [3], [4].
> > 3. The Helpsteer dataset also includes preference dimensions we used, such as complexity and verbosity.
> > 4. In the context of this paper, the core of the preference lies in user-specific preference dimensions, such as creativity. For these datasets, we only use their questions, which is more similar to real-world applications where LLMs need to provide answers that align with users' real-time preferences for various questions. Hence, we chose these datasets.
>
>
> **Reference:**
>
> [1] Gao S, Ge Q, Shen W, et al. Linear Alignment: A Closed-form Solution for Aligning Human Preferences without Tuning and Feedback. ICML24. (https://arxiv.org/abs/2401.11458)
>
> [2] Li X, Zhang T, Dubois Y, et al. Alpacaeval: An automatic evaluator of instruction-following models. (https://github.com/tatsu-lab/alpaca_eval)
>
> [3] Guo Y, Cui G, Yuan L, et al. Controllable preference optimization: Toward controllable multi-objective alignment. ACL24. (https://aclanthology.org/2024.emnlp-main.85.pdf)
>
> [4] Yang K, Liu Z, Xie Q, et al. Metaaligner: Towards generalizable multi-objective alignment of language models. NeuIPS24. (https://arxiv.org/abs/2403.17141v2)

---

> ### Author Response · Authors · 2024-11-24
>
> Dear Reviewer VESu,
>
> Thank you for your insightful feedback and valuable comments on our paper. We have carefully addressed the concerns you raised, as outlined in the revised version. Could you kindly confirm if the revisions adequately address the issues? If there are any remaining points that require further discussion or clarification, we would be more than willing to engage and improve the work accordingly.
>
> Additionally, if you find the changes satisfactory, we would appreciate your consideration for adjusting the score to reflect the improvements.
>
> Looking forward to your feedback.

---

> ### Author Response · Authors · 2024-11-28
> **Supplementary experiment for Q6**
>
> Thank you for raising the question of whether our method is effective across LLMs of different sizes. Although we have presented the results using the reward model score in A6, considering your request for more evaluation metrics in Q4, we aim to provide a more comprehensive demonstration of Amulet's performance. Therefore, we have supplemented our analysis with the GPT-4o win rate as a metric. Due to space limitations, we only present the average results for different models and preference dimensions (more details for each baseline are also presented in the anonymous link: https://anonymous.4open.science/r/amulet-detailed-experiment-results-2E46/README.md), as shown in the two tables below:
>
> **Amulet vs. all the baselines on different LLMs**
> |                                |   win_ratio |   tie_ratio |   lose_ratio |
> |:-------------------------------|------------:|------------:|-------------:|
> | Qwen2-0.5B-Instruct     |     56.160   |     8.623 |      35.217 |
> | Llama-3.2-1B-Instruct |     77.063 |     7.132 |      15.805  |
> | Llama-2-13B-chat     |     83.121 |     3.878  |      13.001 |
> | Llama-2-70B-chat     |     81.492 |     4.485   |      14.022 |
>
> **Amulet vs. all the baselines on different preference dimensions**
> |                  |   win_ratio |   tie_ratio |   lose_ratio |
> |:-----------------|------------:|------------:|-------------:|
> | concise   |     65.915  |    7.725    |      26.359 |
> | creative  |     80.839 |    9.063  |      10.097 |
> | uplifting |     79.137 |    6.493  |      14.369 |
> | verbose   |     71.944 |    0.836 |      27.219 |
>
> We use the GPT-4o win rate as the evaluation metric in this experiment in the same way as we did in A4. As shown in these two tables and the anonymous link, except for only one setting of Qwen2-0.5B-Instruct under verbose preference, Amulet achieved the highest win rate in all other settings. Moreover, compared to the results using the reward model score shown in A6, Amulet shows a better improvement in achieving the best average performance.

---

### Official Review · Reviewer_kahG · 2024-12-12

**Soundness:** 3
**Presentation:** 3
**Contribution:** 3
**Rating:** 6
**Confidence:** 3

**Summary:**

This work presents Amulet, a new framework to adapt language models to individual user preferences at test time. It treats the decoding of each token as a separate online learning problem, guided by both a base prompt and a preference prompt. It also provides a closed-form solution for each iteration step of the optimization process, making the method pretty computational efficient. Experimental results demonstrate the superiority of Amulet across various combinations of LLMs, datasets, and user preferences.

**Strengths:**

S1: Test-time realignment for personalized preferences is an interesting research topic in the field of LLMs.

S2: The proposed framework approaches the decoding of each token as an independent online learning problem and introduces a closed-form solution for optimization, which is novel to me.

S3: The paper is well-written, and the experimental results seems good.

**Weaknesses:**

W1: Insufficient baselines. Only Pref and LA are included for comparison. More alignment approaches are needed. Additionally, the Pref baseline appears trivial in its implementation. Given that the studied preference dimensions in this work are easy to define, it can be effective to use more sophisticated prompt engineering approaches that could serve as stronger baselines, e.g., emphasizing the output format.

W2: Lack of Evaluation on Implicit Preferences. While the paper demonstrates effectiveness on eight explicit preference dimensions (creative, verbose, concise, etc.), it is unknown whether the proposed method can solve the scenarios with implicit user preference.

W3: Dataset Relevance. Three out of four used datasets (HelpSteer, Truthful QA, and UltraChat) appear to have limited relevance to the task of aligning with user preferences.

**Questions:**

See the weaknesses above.

---

### Author Response · Authors · 2024-11-22
**Revised Manuscript**

Dear Reviewers,

Thank you once again for your valuable comments and suggestions on our paper. We have carefully revised our manuscript and conducted additional experiments based on your feedback. For the reviewers’ convenience, **we have highlighted some of the significant changes in the revised manuscript in red**. Below is a summary of the modifications and updates made:
1. **Ablation Setups** (@Reviewer VESu): We update Section 4.3 to include the tuning range for each hyperparameter. Appendix B.4 provides additional details on the ablation experiments.
3. **Amulet variants (varied $\alpha$)** (@Reviewer VESu): For the experiment where all other parameters remain fixed and only $\alpha$ is varied, the setup is detailed in Section 4.3. Additional results on the performance of different Amulet variants formed by adjusting $\alpha$-$\lambda$ pairs are provided in Appendix B.4.
4. **Clarification of "real-time"** (@Reviewer VESu, dMQ4): We supplement our "real-time" setting in Section 3.3 (lines 232-235) and emphasize its distinction from recommendation systems.
5. **Performance on GPT-4o Win Rate** (@Reviewer VESu): We add the experimental results and analysis using the GPT-4o win rate metric in Section 4.2, with the findings visualized in Figure 3. Further details on the GPT-4o win rate experiments are provided in Appendix B.2 due to space limitations in the main text.
6. **Effect of Model Size** (@Reviewer VESu): We analyze the influence of varying model sizes on Amulet’s performance in Section 4.3, with the experimental results summarized in Table 2 for clarity.
7. **Dataset Selection** (@Reviewer VESu): We expand Section 4.1 to provide a more detailed explanation of how these datasets are utilized and included a brief justification for their selection.
8. **More Baselines** (@Reviewer KgK2, dMQ4): In Appendix B.3, we include additional baselines, such as beam search with beam numbers 4, 8, and 16, as well as RAIN [1]. Our results show that beam search performs poorly, while RAIN incurs an extraordinarily high computational cost, **taking on average 17.1 times more time per token**！
9. **Time Efficiency** (@Reviewer KgK2): We provide a brief introduction in lines 270–272 and include Appendix B.5 for a detailed analysis of the computational efficiency of Amulet compared to other baseline algorithms.

We hope these updates adequately address your concerns and demonstrate the robustness of our approach. If there are any additional issues or points requiring clarification, please feel free to let us know.


**Reference**

[1] Li Y, Wei F, Zhao J, et al. RAIN: Your Language Models Can Align Themselves without Finetuning. ICLR24.

---

### Meta-Review · Area_Chair_5HT4 · 2024-12-21

**Metareview:**

In this paper, the authors propose a test-time realignment framework for personalized preference adaptation. Reviewers found the paper well-written and the topic highly interesting. However, they expressed concerns about the sufficiency of the baselines and the relevance of the dataset to the topic. The authors provided satisfactory answers in the rebuttal, and overall, I believe this work is above the acceptance threshold.

**Additional Comments On Reviewer Discussion:**

In the rebuttal, the authors clarified their experimental and dataset settings, conducted additional experiments with different model sizes, and reported the computational costs. One reviewer agreed that these responses addressed all concerns, while the other reviewers did not provide further feedback.

---

### Decision · Program_Chairs · 2025-01-22

Accept (Poster)